# Label consistency in overfitted generalized $k$-means

**Linfan Zhang**
Department of Statistics
University of California, Los Angeles
Los Angeles, CA 90095
linfanz@g.ucla.edu

**Arash A. Amini**
Department of Statistics
University of California, Los Angeles
Los Angeles, CA 90095
aaamini@ucla.edu

## Abstract

We provide theoretical guarantees for label consistency in generalized $k$-means problems, with an emphasis on the overfitted case where the number of clusters used by the algorithm is more than the ground truth. We provide conditions under which the estimated labels are close to a refinement of the true cluster labels. We consider both exact and approximate recovery of the labels. Our results hold for any constant-factor approximation to the $k$-means problem. The results are also model-free and only based on bounds on the maximum or average distance of the data points to the true cluster centers. These centers themselves are loosely defined and can be taken to be any set of points for which the aforementioned distances can be controlled. We show the usefulness of the results with applications to some manifold clustering problems.

## 1 Introduction

Consider the problem of clustering data points sampled according to some probability measure $\mu$ from a normed space $\mathcal{X}$ with norm $\| \cdot \|_{\mathcal{X}}$. In the ideal setting, the generalized $k$-means clustering minimizes the population cost function

$$Q(\xi; \mu) := \left( \int \min_{1 \leq \ell \leq L} \|x - \xi_\ell\|_{\mathcal{X}}^p d\mu(x) \right)^{1/p} \tag{1}$$

where $\xi = (\xi_1, \ldots, \xi_L) \in \mathcal{X}^L$ is a set of $L$ vectors in $\mathcal{X}$, for some fixed integer $L$. In practical data analysis, we are given a sample $\{x_1, \ldots, x_n\}$ drawn from $\mu$ and solve an empirical version of (1), namely,

$$\widehat{Q}(\xi) = Q(\xi; \mathbb{P}_n) := \left( \frac{1}{n} \sum_{i=1}^n \min_{1 \leq \ell \leq L} \|x_i - \xi_\ell\|_{\mathcal{X}}^p \right)^{1/p}. \tag{2}$$

Here, $\mathbb{P}_n := \frac{1}{n} \sum_{i=1}^n \delta_{x_i}$ is the empirical measure associated with the sample and $\delta_x$ is the point mass measure at $x$. The minimizer of $\widehat{Q}(\cdot)$ over $\mathcal{X}^L$ is denoted as $\widehat{\xi} = (\widehat{\xi}_1, \ldots, \widehat{\xi}_L)$ and each point $x_i$ is assigned a cluster label $\widehat{z}_i := \operatorname{argmin}_\ell \|x_i - \widehat{\xi}_\ell\|_{\mathcal{X}}$.

Meanwhile, we assume that each data point $x_i$ also has a true cluster label $z_i \in [K] := \{1, \ldots, K\}$ which is determined solely by an unknown data-generating process. These true labels are not necessarily related to the optimal solutions of (1) or (2). To distinguish the two, we refer to the clustering induced by $(z_i)$ as the **true clustering**, while a clustering that minimizes the generalized $k$-means cost function (2), i.e., the clustering induced by $(\widehat{z}_i)$, is referred to as an **optimal $k$-means clustering**. In this paper, we consider the *label consistency* problem, that is, how close the labels $(\widehat{z}_i)$ estimated by $k$-means clustering are to the true labels $(z_i)$. Note that we allow the number of $k$-means clusters $L$ to be different from the true number of clusters $K$.

35th Conference on Neural Information Processing Systems (NeurIPS 2021).

In the above formulation, the case where $p = 2$, $\mathcal{X} = \mathbb{R}^d$ and $\|\cdot\|_{\mathcal{X}}$ is the Euclidean norm leads to the classical and widely used $k$-means problem. Much of the theoretical analysis of $k$-means has been performed in this case. Early work has focused on how close the optimization problems based on the empirical and ideal cost functions (2) and (1) are to each other, where the closeness is measured in terms of the recovered centers (i.e., $\widehat{\xi}$ and $\xi$) or the optimal value of the objective function.

Such consistency results are proved, for the global minimizers of (2), in the early work of [22, 29] and also in [30, 19] from the vector quantization perspective. These classical results do not directly apply to the performance of the $k$-means in practice, mainly because solving (2) is NP-hard and approximation methods are usually applied to deal with it. Also, considerations of the label consistency problem are absent from this line of work since no true clustering, external to the $k$-means problem, is assumed to exist.

More recently, there has been more interest in the consistency of practical $k$-means algorithms [14, 21] as well as the label consistency problem. Lu and Zhou [21] obtain sharp bounds on the label consistency of the Lloyd's algorithm [20] under a sub-Gaussian mixture model. Semidefinite programming (SDP) relaxation is another popular technique for deriving polynomial-time approximations to the $k$-means problem [28]. Its label consistency has been studied when data is generated from the stochastic ball model [4, 10], sub-Gaussian mixtures [25, 8, 9], the Stochastic Block Model (SBM) [9] and general models [18]. Convex clustering is another relaxation method whose label consistency has been discussed in [34, 27, 11, 31]. The literature on community detection in SBM, a network clustering problem, is also mainly focused on label consistency and inspires our work here; see [1, 33] for a review of those results. For label consistency in kernel spectral clustering, see [2].

In this paper, we study the label consistency of approximate solutions of the generalized $k$-means problem (2) when $L \geq K$. Our focus will be on the overfitted case where $L > K$. This is often relevant in practice since the data-generating process may have a natural number of clusters $K$ that is unknown a priori. An example is the sub-Gaussian mixture with $K$ components. More interesting examples are given in Section 3. All the aforementioned works on label consistency exclusively consider the correctly-fitted case $L = K$. We show that when the data-generating process admits a set of centers that satisfy certain separation conditions, estimated labels with $L \geq K$ clusters, are close to a *refinement* of the true labels. These bounds reduce to the label consistency criteria for $L = K$, but have no counterpart in the literature for $L > K$.

Overfitting in $k$-means is considered in [32, 23] where it is shown to improve the approximation factor (see Assumption 1(b)) of certain polynomial-time $k$-means algorithms. Analysis of the approximation factor is concerned with how close one can get to the optimal value of the $k$-means objective function. In contrast, we are concerned with the label recovering problem and not directly concerned with how well the objective function is approximated. Our work is also aligned with the recent trend of *beyond worst case* analysis of the NP-hard problems [6], where the performance of the algorithms are considered assuming that there are some meaningful structures in the data (e.g., true clusters). We refer to Remark 1 for a more detailed comparison with this literature.

Our results are algorithm-free in the sense that they apply to any algorithm that achieves a constant-factor approximation to the optimal objective. They are also model-free in the sense that we do not make any explicit assumption on the data-generating process. This is important in practice, since many common data models, such as sub-Gaussian mixtures, are often too simplified to capture real clustering problems. We provide examples of more complicated data models in Section 3 and show how our general results can provide new insights for these models. Since $k$-means clustering often appears as a building block in many sophisticated clustering algorithms, we believe our results will be of broad interest in understanding the performance of clustering algorithms in overfitted settings.

**Notation.** $Q(\xi; \mu)$ is only dependent on the set of values among the coordinates of $\xi$. Although we view $\xi$ as a vector (for which the order of elements matter), with some abuse of notation, we view $Q(\cdot; \mu)$ as a set function (mapping $2^{\mathcal{X}}$ to $\mathbb{R}$) that is only sensitive to the set of values represented by $\xi$. This justifies using the the same symbol for the function irrespective of the number of coordinates of $\xi$, i.e., the number of clusters. The reason to keep $\xi$ as an (ordered) vector is to make the cluster labels well-defined. For simplicity, let $\|\cdot\| = \|\cdot\|_{\mathcal{X}}$. For the case where $\mathcal{X} \subset \mathbb{R}^d$, one often takes $\|\cdot\|$ to be the Euclidean norm, but our results are valid for any norm on $\mathbb{R}^d$, and more broadly any normed space $\mathcal{X}$.

## 2 Main Results

We first state assumptions about the $k$-means clustering algorithm.

**Assumption 1.** *Consider an algorithm for the generalized $k$-means problem* (2), *referred to as ALG(p) hereafter, and let $\widehat{\xi}^{(L)} \in \mathcal{X}^L$ and $\widehat{\xi}^{(K)} \in \mathcal{X}^K$ be its estimated centers when applied with $L$ and $K$ clusters, respectively. Let $L \geq K$. Assume that ALG(p) has the following properties, for all input sequences $(x_i)$:*

*(a)* *Efficiency: The Voronoi cell of every estimated center $\widehat{\xi}_\ell^{(L)}$ contains at least one of $(x_i)$.*

*(b)* $\kappa$-*approximation:* $\widehat{Q}(\widehat{\xi}^{(K)}) \leq \kappa \cdot \min_{\xi \in \mathcal{X}^K} \widehat{Q}(\xi)$, *and similarly with $K$ replaced by $L$.*

Efficiency can be achieved by substituting centers whose Voronoi cells have an empty intersection with $\{x_i\}$, by those having the opposite property. For $\kappa$-approximation, the factor $\kappa$ can depend on the number of clusters $K$ (or $L$). For example, the $k$-means++ algorithm has $\kappa = O(\log K)$, with high probability over the initialization [3]. However, there are also constant-factor approximation algorithms for $k$-means where $\kappa = O(1)$ independent of $K$ (or $L$) [24, 13, 15]. For example, with local search, $k$-means++ can achieve a constant-factor approximation [16]. In addition, $\kappa$-approximation is not required for all inputs. That is, we are not concerned with the worst-case approximation factor. The $\kappa$ in Assumption 1(b) is the approximation factor of the algorithm on the specific data under consideration. It is enough for an algorithm to achieve good approximation only on the data of interest.

For some of the results, Assumption 1(b) can be replaced with the following modified version: $(b')$ $\kappa$-approximation only for $K$ clusters plus a mononoticity assumption: $\widehat{Q}(\widehat{\xi}^{(L)}) \leq \widehat{Q}(\widehat{\xi}^{(K)})$. Mononoticity is also a reasonable requirement and obviously true for the exact $k$-means solutions.

Next, we extend the definition of the misclassification rate to the overfitted case.

**Definition 1.** The (generalized) misclassification rate between two label vectors $z \in [K]^n$ and $\widehat{z} \in [L]^n$, with $K \leq L$, is

$$\text{Miss}(z, \widehat{z}) = \min_\omega \frac{1}{n} \sum_{i=1}^n 1\{z_i \neq \omega(\widehat{z}_i)\},$$

where the minimization ranges over all surjective maps $\omega : [L] \to [K]$.

When $L = K$, a surjective map $\omega$ is necessarily a bijection and the above becomes the usual definition of misclassification rate when the number of clusters is correctly identified. In this case, $\text{Miss}(z, \widehat{z}) = 0$ means that there is a one-to-one correspondence between the estimated and true clusters. The generalized definition above allows us to extend this notion of exact recovery to the case $L > K$. In particular, $\text{Miss}(z, \widehat{z}) = 0$ when $L > K$, if and only if $\widehat{z}$ is a *refinement* of $z$. To see this, note that $\text{Miss}(z, \widehat{z}) = 0$ implies the existence of a map $\omega : [L] \to [K]$ such that $\omega(\widehat{z}_i) = z_i$ for all $i$. This in turn is equivalent to: $\widehat{z}_i = \widehat{z}_{i'} \implies z_i = z_{i'}$, which is the refinement relation for the associated clusters. In general, $\text{Miss}(z, \widehat{z})$ is small if $\widehat{z}$ is close to a refinement of $z$.

We also use the (optimal) matching distances between elements of two vectors viewed as sets.

**Definition 2.** For $\xi \in \mathcal{X}^L$ and $\xi^* \in \mathcal{X}^K$, define the $\ell_\infty$ and $\ell_p$ optimal matching distances as

$$d_\infty(\xi, \xi^*) = \min_\sigma \max_{1 \leq k \leq K} \|\xi_{\sigma(k)} - \xi_k^*\|, \quad d_p(\xi, \xi^*) = \min_\sigma \Big( \sum_{k=1}^K \|\xi_{\sigma(k)} - \xi_k^*\|^p \Big)^{1/p},$$

where $\sigma : [K] \to [L]$ ranges over all injective maps.

For $K = L$, $d_\infty$ is an upper bound on the Hausdorff distance between the two sets. Obviously, we have $d_\infty \leq d_p$ for any $p \geq 1$.

### 2.1 Distance to true centers

Let $z = (z_i)_{i=1}^n \in [K]^n$ be a given set of true labels for the data points $(x_i)_{i=1}^n$. In addition, our results are stated in terms of a set of vectors $\xi^* = (\xi_k^*)_{k=1}^K$ which we refer to as the "true centers".

Throughout, $\xi^*$ will be only vaguely specified. The only requirement on $\xi^*$ is that together with the observed data points $(x_i)$ and the true labels $(z_i)$, they satisfy the deviation bounds in each theorem, e.g., $\max_{1 \le i \le n} \|x_i - \xi^*_{z_i}\| \le \eta$ in Theorem 1, etc. Let $\pi_k = \sum_{i=1}^n 1\{z_i = k\}/n$ be the proportion of observed data points in true cluster $k$ and let $\pi_{\min} = \min_k \pi_k$.

We let $\widehat{\xi}$ be a solution of the $k$-means algorithm with $L \ge K$ centers and let $\widehat{z}_i \in \operatorname{argmin}_\ell \|x_i - \widehat{\xi}_\ell\|$ be the corresponding estimated labels. Our first result provides guarantees for exact label recovery, in the extended sense of recovering a refinement of the true partition when $L > K$ and the exact partition when $L = K$.

**Theorem 1** (Exact recovery). *Consider a vector of (true) centers $\xi^* \in \mathcal{X}^K$ and labels $(z_i)_{i=1}^n \in [K]^n$. Pick $\eta, \delta > 0$ such that $\max_{1 \le i \le n} \|x_i - \xi^*_{z_i}\| \le \eta$, and*

$$\min_{(k,k'): k \ne k'} \|\xi^*_k - \xi^*_{k'}\| \ge \delta. \tag{3}$$

*Consider an algorithm ALG(p) for problem* (2), *satisfying Assumption 1, and let $(\widehat{z}_i)_{i=1}^n \in [L]^n$ and $\widehat{\xi} \in \mathcal{X}^L$ be the estimated labels and centers of ALG(p) applied with the $L \ge K$. Then,*

$$\frac{\delta}{\eta} > 2\frac{(1+\kappa)}{\pi_{\min}^{1/p}} + 4 \quad \implies \quad \operatorname{Miss}(z, \widehat{z}) = 0, \quad d_p(\widehat{\xi}, \xi^*) \le \frac{(1+\kappa)\eta}{\pi_{\min}^{1/p}}. \tag{4}$$

When $L = K$, the assertion $\operatorname{Miss} = 0$ means that there is a permutation $\sigma$ on $[K]$ such that $\sigma(\widehat{z}_i) = z_i$ for all $i$, that is, we have the exact recovery of labels $(z_i)$ in the classical sense. When $L > K$, Theorem 1 guarantees the exact recovery of a refinement of the true labels $(z_i)$.

**Example 1** (Stochastic Ball Model). Assume that data are generated from the stochastic ball model considered in [26], where $x_i = \xi^*_{z_i} + r_i$ with $r_i$ sampled independently from a distribution supported on the unit ball in $\mathbb{R}^d$. Here, $\{\xi^*_k\}_{k=1}^K \subset \mathbb{R}^d$ are a fixed set of centers. Clearly, we can take $\eta = 1$ in Theorem 1. Then, any $\kappa$-approximate $k$-means algorithm achieves exact recovery when $\delta > 2 + 2(1+\kappa)/\sqrt{\pi_{\min}}$ for $L = K$. In the overfitted case, when $\delta > 4 + 2(1+\kappa)/\sqrt{\pi_{\min}}$, the estimated label vector is an exact refinement of the true labels $(z_i)$. $\square$

In the above example, although it is intuitively clear that for a sufficiently large $\delta$, the solution of the $k$-means problem should achieve exact label recovery (in the extended sense), Theorem 1 allows us to provide a provable guarantee for any constant-factor approximation, with an explicit bound on the separation parameter $\delta$.

We now turn to approximate recovery where the misclassification rate is small.

**Theorem 2** (Approximate Recovery). *Consider a vector of (true) centers $\xi^* \in \mathcal{X}^K$ and labels $(z_i)_{i=1}^n \in [K]^n$. Pick $\varepsilon, \delta > 0$ such that $(\frac{1}{n} \sum_{i=1}^n \|x_i - \xi^*_{z_i}\|^p)^{1/p} \le \varepsilon$, and (3) holds. Consider an algorithm ALG(p) for problem* (2), *satisfying Assumption 1, and let $(\widehat{z}_i)_{i=1}^n \in [L]^n$ and $\widehat{\xi} \in \mathcal{X}^L$ be the estimated labels and centers of ALG applied with the $L \ge K$. Then, for any $c > 2$,*

$$\frac{\delta}{\varepsilon} > \frac{(1+\kappa)c}{\pi_{\min}^{1/p}} \quad \implies \quad \operatorname{Miss}(z, \widehat{z}) < K(1+\kappa)^p c^p \left(\frac{\varepsilon}{\delta}\right)^p, \quad d_p(\widehat{\xi}, \xi^*) \le \frac{(1+\kappa)\varepsilon}{\pi_{\min}^{1/p}}. \tag{5}$$

The key difference between Theorems 1 and 2 is the bounds assumed on the deviations $D_i := \|x_i - \xi^*_{z_i}\|, i \in [n]$. Theorem 1 assumes a bound on the maximum distance to true centers, $\max_i D_i$, while Theorem 2 assumes a bound on an average distance, $(\frac{1}{n} \sum_i D_i^p)^{1/p}$, leading to a more relaxed condition.

Theorem 2 provides an upper bound on the misclassification rate when a certain separation condition is satisfied. To simplify, consider the case $K = \kappa = p = 2$ and take $c = 2.1$. Then, Theorem 2 implies the following: For every $\beta > 0$, there exists a constant $c_1(\beta, \pi_{\min}) > 0$ such that if

$$\delta/\varepsilon \ge c_1(\beta, \pi_{\min}), \tag{6}$$

then any 2-factor $k$-means algorithm will have $\operatorname{Miss} \le \beta$ to the target labels. The next proposition shows that condition (6) is sharp up to constants.

**Proposition 1.** *There exists a family of datasets $\{(x_i, z_i)\}_{i=1}^n$, with $K = 2$ balanced true clusters (i.e., $\pi_{\min} = 1/2$) and parameterized by true center separation $\delta$ and $\varepsilon = (\frac{1}{n} \sum_{i=1}^n \|x_i - \xi^*_{z_i}\|^2)^{1/2}$*

*with the following property: Given any constant $\beta \in (0, 1/2)$, there exists a constant $c_2(\beta) > 0$, such that if $\delta/\varepsilon < c_2(\beta)$, then any 2-factor k-means approximation algorithm with $L = 2$ clusters has misclassification rate satisfying $\frac{1}{2} - \beta \leq Miss \leq \frac{1}{2}$. Moreover, any 2-factor k-means approximation algorithm with $L = 4$ clusters will recover a perfect refinement of the original clusters in the above setting.*

The proof of Proposition 1 can be found in the Supplementary Material. This proposition shows that if the separation condition (6) is reversed, one can force the performance of any $k$-means algorithm to be arbitrarily close to that of random guessing. The true centers in Proposition 1 are the natural centers implied by the $k$-means cost function for the true labels, that is, $\xi_k^* = \frac{1}{n} \sum_i x_i 1\{z_i = k\}$ for $k = 1, 2$. One can take $c_1(\beta, \pi_{\min}) = 6.3 \max(1/\pi_{\min}, 2/\beta)^{1/2}$ and $c_2(\beta) = \sin(\tan^{-1}(\sqrt{\beta/45}))$ for the constants in (6) and Proposition 1.

**Remark 1.** The separation condition (6) is related to the *distribution stability* introduced in [5]. Roughly speaking distribution stability plus the following property implies our condition:

(D1) For every pair of distinct clusters $C_k$ and $C_\ell$ with centers $\xi_k^*$ and $\xi_\ell^*$, there is a point $x \in C_\ell$ such that $\|x - \xi_k^*\| \leq \|\xi_\ell^* - \xi_k^*\|$.

That is, every cluster $C_\ell$ has points which are closer than $\xi_\ell^*$ to the centers of other clusters. This property is quite mild and one expects it to hold with high probability if the distribution of the points have positive density w.r.t. to the (full-dimensional) Lebesgue measure in a ball around the center. The above seems to suggest that distribution stability is slightly weaker than our condition (6). However, in the presence of (D1), we can also significantly relax distribution stability to arrive at our condition, the details of which are provided in the Supplementary Material. In this sense, these two notions are closely related but not directly comparable, i.e., neither is weaker than the other in general.

**Example 2** (Sub-Gaussian mixtures). Let us assume that the data is generated from a $K$-component sub-Gaussian mixture model $x_i = \xi_{z_i}^* + d^{-1/2} w_i$ where $w_i = (w_{i1}, \ldots, w_{id}) \in \mathbb{R}^d$ is a zero mean sub-Gaussian noise vector with sub-Gaussian parameter $\sigma_i$, and $z_i \in [K]$ is the latent label of the $i$th observation. This is an extension of the sub-Gaussian mixture model considered in [7]. Determining whether $(\xi_k^*)_{k=1}^K$ is actually the solution of the population problem (1) is, itself, challenging and the answer may depend on the exact distribution of $\{w_i\}$. Nevertheless, our results allow us to treat $(\xi_k^*)$ as the true centers. Below we sketch how Theorem 2 applies in this case. The details of the arguments, including the exact definition of a sub-Gaussian vector are provided in the Supplementary Material. Let $\sigma_{\max} = \max_i \sigma_i$ and set $\alpha_i^2 := \mathbb{E}\|d^{-1/2} w_i\|_2^2$ and $\bar{\alpha}_n^2 := \frac{1}{n}\sum_{i=1}^n \alpha_i^2$. Assume that there is a numerical constant $C > 0$ such that $\bar{\alpha}_n^2 \leq C\sigma_{\max}^2$. Then, one can show that

$$\mathbb{P}\Big(\frac{1}{n}\sum_{i=1}^n \|x_i - \xi_{z_i}^*\|^2 > 2\bar{\alpha}_n^2\Big) \leq e^{-c_1 n \bar{\alpha}_n^4/\sigma_{\max}^4} =: p_n$$

for some numerical constant $c_1 > 0$. Taking $\varepsilon^2 = 2\bar{\alpha}_n^2$ and $p = 2$ in Theorem 2, we have that with probability at least $1 - p_n$,

$$\frac{\delta^2}{2\bar{\alpha}_n^2} > \frac{(1+\kappa)^2 c^2}{\pi_{\min}} \quad \implies \quad \text{Miss}(z, \hat{z}) \leq 2K(1+\kappa)^2 c^2 \Big(\frac{\bar{\alpha}_n}{\delta}\Big)^2,$$

where $\delta$ is as in (3) and $c > 2$. In a general problem, $\bar{\alpha}_n$, $\sigma_{\max}$ and $\delta$ all can vary with $n$. In order to have label consistency for an ALG(2) algorithm, it is enough to have $\bar{\alpha}_n/\delta = o(1)$ and $n\bar{\alpha}_n^4/\sigma_{\max}^4 \to \infty$. The consistency here is based on the extended Definition 1 and includes the overfitted case in which a refinement of the true labels is consistently recovered. We note that the model in this example includes a very general Gaussian mixture model as a special case, namely the case $w_i \sim N(0, \Sigma_i)$ where the covariance matrices $\Sigma_i \in \mathbb{R}^{d \times d}$ are allowed to vary with each data point. In this case, one can take $\sigma_{\max} = \max_{1 \leq i \leq n} \|\Sigma_i\|_{\text{op}}$ where $\|\cdot\|_{\text{op}}$ denotes the operator norm, and $\bar{\alpha}_n^2 := \frac{1}{n}\sum_{i=1}^n \text{tr}(\Sigma_i)/d$. ☐

## 2.2 Distance to fake centers

We now extend Theorem 2, to allow for "fake" centers $\{\widetilde{\xi}_\ell\}_{\ell=1}^L$ and the corresponding labels $\{\widetilde{z}_i\}$. These can be constructed to reduce the required distance to the data points $(x_i)$.

**Theorem 3** (Approximate Recovery, II). *For a fixed $L \geq K$, consider a vector of constructed centers $\widetilde{\xi} \in \mathcal{X}^L$, constructed labels $\widetilde{z} = (\widetilde{z}_i)_{i=1}^n \in [L]^n$ and true labels $z = (z_i)_{i=1}^n \in [K]^n$. Assume that $\widetilde{z}$ is a refinement of $z$, i.e. there is $\widetilde{\omega} : [L] \to [K]$ such that $\widetilde{\omega}(\widetilde{z}_i) = z_i$ for all $i \in [n]$. Pick $\varepsilon, \delta > 0$ such that*

$$\left( \frac{1}{n} \sum_{i=1}^n \|x_i - \widetilde{\xi}_{\widetilde{z}_i}\|^p \right)^{1/p} \leq \varepsilon, \quad \min_{\ell_1 \neq \ell_2, \, \widetilde{\omega}(\ell_1) \neq \widetilde{\omega}(\ell_2)} \|\widetilde{\xi}_{\ell_1} - \widetilde{\xi}_{\ell_2}\| \geq \delta \tag{7}$$

*Consider an algorithm ALG(p) for problem (2), satisfying Assumption 1, and let $(\widehat{z}_i)_{i=1}^n \in [L]^n$ be the estimated label vector of ALG(p) applied with $L$ clusters. Then, for any $c > 2$,*

$$\frac{\delta}{\varepsilon} > \frac{(1+\kappa)c}{\pi_{\min}^{1/p}} \quad \implies \quad \mathrm{Miss}(z, \widehat{z}) < K(1+\kappa)^p c^p \left( \frac{\varepsilon}{\delta} \right)^p. \tag{8}$$

The advantage of Theorem 3 is that when the desired number of clusters $L$ increases, the bound on the misclassification rate can go down: In some applications, by carefully constructing the fake centers $\widetilde{\xi}$, we can make $\varepsilon$ smaller as $L$ increases, while roughly maintaining the minimum separation among fake centers associated with the true clusters. If successful, this implies that a refinement of the true clustering can be achieved even when it is hard to recover the true clustering itself. In the following section, we show how this strategy can be applied to some manifold clustering problems.

## 3 The case for overfitting

We now illustrate applications of Theorem 3 in settings where it is hard to recover true clusters, based on the ideal $K$, but it is possible to obtain accurate refinements by overfitting. The idea is to consider clusters that look like submanifolds of $\mathbb{R}^d$.

### 3.1 Mixture of curves

We say that a random variable $x$ has a $(\rho, \sigma)$ *sub-Gaussian curve distribution* if $x = \gamma(t)$ where $t \in \mathbb{R}$ has a sub-Gaussian distribution with parameter $\sigma$ and $\gamma : \mathbb{R} \to \mathbb{R}^d$ is a locally $\rho$-Lipschitz map. i.e., $\|\gamma(t) - \gamma(s)\| \leq \rho |t - s|$ for all $t, s \in \mathbb{R}$ such that $|t - s| \leq \frac{1}{\rho}$.

**Proposition 2.** *Assume that $(x_i)_{i=1}^n$ are independent draws from a $K$-component mixture of $(\rho, \sigma)$ sub-Gaussian curve distributions. That is, $x_i = \gamma_{z_i}(t_i)$ where $z_i \in [K]$, $t_i \sim \mathbb{Q}_{z_i}$ independently across $i$, each $\mathbb{Q}_k$ is a sub-Gaussian distribution on $\mathbb{R}$ with parameter $\sigma$, and each $\gamma_k$ is locally $\rho$-Lipschitz. Let $\mathcal{C}_k$ be the support of the distribution of $\gamma_k(t)$ where $t \sim \mathbb{Q}_k$. Assume that*

$$\min_{x \in \mathcal{C}_k, \, y \in \mathcal{C}_{k'}} \|x - y\| \geq \delta > 0, \quad \text{for all } k \neq k'.$$

*Then, there exist a constant $C = C(K, \delta, \rho, \sigma, \kappa)$ such that any ALG(2) satisfying Assumption 1 applied with $L_n \leq C\sqrt{n \log n}$ clusters recovers a perfect refinement of $z$ with probability $\geq 1 - n^{-1}$.*

The significance of this result is that one recovers a perfect refinement with the number of partitions $L_n = o(n)$. It is trivial to obtain a perfect refinement with $L_n = n$, but not so with $L_n/n \to 0$. This is especially the case since one can achieve quite complex cluster configurations with the model in Proposition 2, for some of which applying $k$-means with $K$ clusters will have a misclassification rate bounded away from zero. Section 3.3 provides some such examples where the true cluster centers coincide, causing any $k$-means algorithm applied with the true $K$ to incur a substantial error. See also Supplementary Material for a discussion of whether $L_n = O(\sqrt{n \log n})$ can be improved.

Various extensions of Proposition 2 are possible. We have the following extension to the noisy setting.

**Corollary 1.** *Assume that the data is given by $y_i = x_i + \frac{1}{\sqrt{d}} w_i$ for $i \in [n]$ where $(x_i)$ are as given in Proposition 2 and $w_i$ are sub-Gaussian noise vectors as in Example 2. Then, under the same assumptions as in Proposition 2, ALG(2) applied with $L_n \leq C\sqrt{n \log n}$ achieves a misclassification rate $\lesssim K(\bar{\alpha}_n/\delta)^2 + \frac{1}{n}$ with probability $\geq 1 - p_n - n^{-1}$ where $\bar{\alpha}_n$ and $p_n$ are defined in Example 2.*

Corollary 1 shows that one can achieve consistent clustering (in the generalized sense) with $L_n = o(n)$ clusters assuming that the noise-to-signal ratio $\bar{\alpha}_n/\delta \to 0$ and $n\bar{\alpha}_n^4/\sigma_{\max}^4 \to \infty$; the same conditions needed in the sub-Gaussian mixture example. Again, this result is significant since even in the noiseless case ($\bar{\alpha}_n = 0$), consistent recovery is not possible with $L = K$ for some mixtures of curve models.

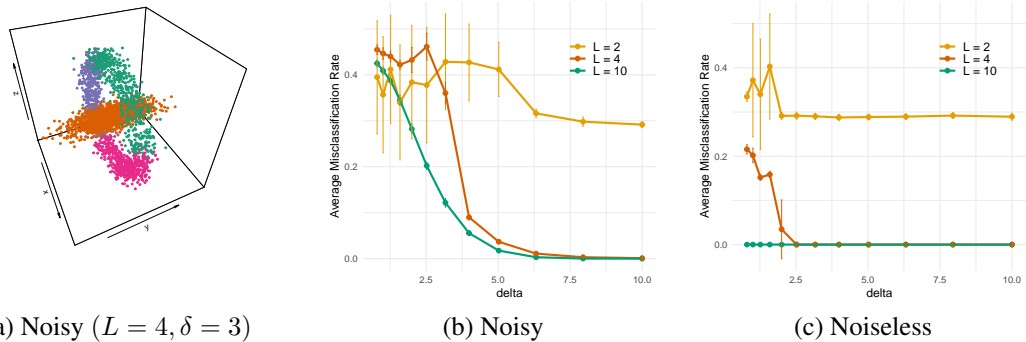

| | |
|---|---|
| (a) Noisy ($L = 4, \delta = 3$) | (b) Noisy | (c) Noiseless |

Figure 1: Line-circle model: (a) Scatter plot for the noisy version. The colors show the $L = 4$ estimated clusters by $k$-means. (b) and (c) show the (generalized) misclassification rate versus $\delta$, the radius of the circle, in the noisy and noiseless versions of the model.

## 3.2 Mixture of higher-order submanifolds

A version of Proposition 2 can be stated for a higher-dimensional version of the mixture-of-curves model, if we consider generalized $k$-means problems with $p > 2$. We say that a random variable $x$ has a $(\rho, \sigma, r)$ sub-Gaussian manifold distribution if $x = \gamma(t)$ where $t \in \mathbb{R}^r$ has a sub-Gaussian distribution with parameter $\sigma$ and $\gamma : \mathbb{R}^r \to \mathbb{R}^d$ is a locally $\rho$-Lipschitz map. i.e., $\|\gamma(t) - \gamma(s)\| \leq \rho\|t - s\|$ for all $t, s \in \mathbb{R}^r$ such that $\|t - s\| \leq \frac{1}{\rho}$.

**Proposition 3.** *Assume that $(x_i)_{i=1}^n$ are independent draws from a $K$-component mixture of sub-Gaussian manifold distributions, with parameters $(\rho, \sigma, r_k)$ for $k \in [K]$, and let $r = \max_{r \in [K]} r_k$. Let $\mathcal{C}_k$ be the support of the distribution of the $k$th component. Assume that*

$$\min_{x \in \mathcal{C}_k, \, y \in \mathcal{C}_{k'}} \|x - y\| \geq \delta > 0, \quad \text{for all } k \neq k'.$$

*Then, there exist a constant $C = C(K, \delta, \rho, \sigma, r, \kappa)$ such that any $ALG(p)$ satisfying Assumption 1, applied with $L_n \leq C(n^{1/p}\sqrt{\log n})^r$ clusters recovers a perfect refinement of $z$ with probability $\geq 1 - n^{-1}$. In particular, for $p > r$, we have perfect refinement recovery with $L_n = o(n)$ clusters, with high probability.*

It is also possible to extend the results to more general distributions on submanifolds via a notion of stochastic covering numbers. For random vector $x$ with distribution $\mu_{\mathcal{C}}$ on a submanifold $\mathcal{C} \subset \mathbb{R}^d$, let $N_{\mu_{\mathcal{C}}}(\varepsilon)$ be the smallest integer for which, there is a high probability $\varepsilon$-cover of $x$, that is, a finite subset $\mathcal{N} \subset \mathcal{C}$ such that $\mathbb{P}(\min_{y \in \mathcal{N}} \|x - y\| \leq \varepsilon) \geq 1 - n^{-2}$. We state a generalization of Proposition 3 to this setting in the Supplementary Material.

## 3.3 Numerical experiments

We first consider the (noiseless) line-circle model in $\mathbb{R}^3$, an example of mixture-of-curves. This model has two clusters: (1) The uniform distribution on the circumference of a circle in the $xz$-plane, centred at the origin, and (2) the standard Gaussian distribution on the $y$ axis. The minimum separation $\delta$ between the two clusters is the radius of the circle. We also consider the noisy version of this model where we add $N(0, \sigma^2 I_3)$. We sample data points with equal probability from the two clusters. It is nearly impossible for the $k$-means to correctly label these two clusters when $L = 2$, since the centers of the two clusters coincide. Figure 1 shows the scatter plot of the data simulated from the noisy line-circle model, with noise level $\sigma = 0.1$, $n = 3000$ and $\delta = 3$. Here, the noise level is set low for better illustration. Different colors are used to label data points based on the output of $k$-means clustering with $L = 4$, and this demonstrates that each estimated cluster is a subset of a true cluster.

The result aligns with Theorem 3. Although, the true centers coincide (with the origin) when $L = 2$, by increasing $L$, we can create fake centers on the line and the circle to have separation close to $\delta$ and thus get a small missclassification rate. The other two panels in Figure 1 show the average missclassification rate over 32 repetitions versus $\delta$, for both the noiseless and noisy ($\sigma = 1$) line-circle

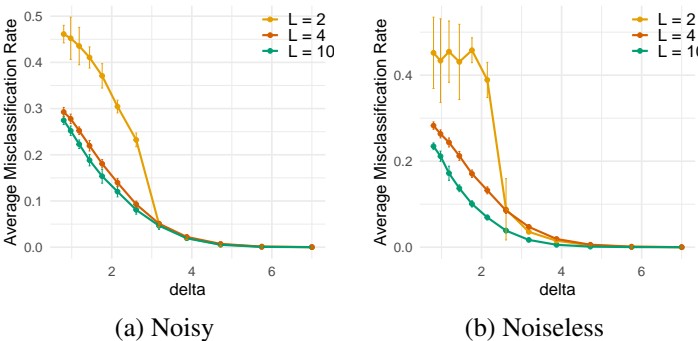

(a) Noisy                  (b) Noiseless

Figure 2: Line-Gaussian model: The (generalized) misclassification rate versus $\delta$, the distance of the Gaussian center to the line, in the (a) noisy and (b) noiseless versions of the model.

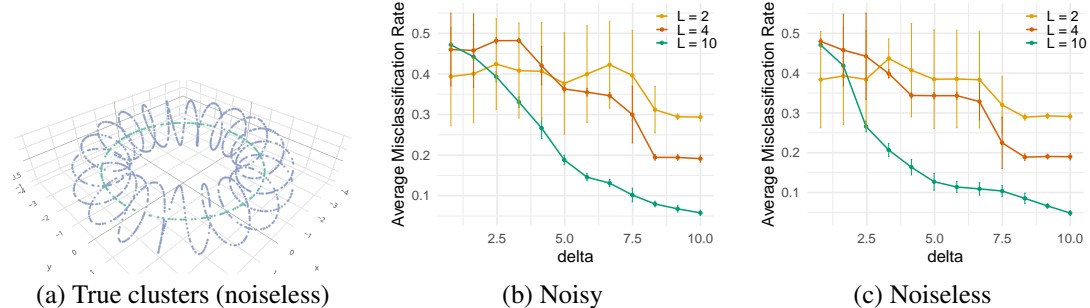

(a) True clusters (noiseless)        (b) Noisy              (c) Noiseless

Figure 3: Circle-torus model: (a) Scatter plot for the noiseless version. Colors are used to separate two true clusters. (b) and (c) show the (generalized) misclassification rate versus $\delta$, the radius of the circle, in the noisy and noiseless versions of the model.

model. Both show that the misclassification rate is negatively associated with $\delta$ and $L$ when $L > 2$. Similar results are shown for the circle-torus model in Figure 3. Details of this model are discussed in the Supplementary Material.

Figure 2 shows the results for a line-Gaussian mixture model: $x_i = \xi^*_{z_i} + \Sigma^{1/2}_{z_i} w_i \in \mathbb{R}^2$ where $\xi^*_1 = (0, \delta)$ and $\xi^*_2 = (0, 0)$, $w_i \sim N(0, I_2)$, $\Sigma_1 = I_2$ and $\Sigma_2 = \mathrm{diag}(\sigma^2, 0)$. Here, we have set $\sigma = 5$ and sampled $n = 3000$ data points with equal probability from the two clusters. We also consider its noisy version by setting all the zero elements in $\Sigma_2$ to 0.7, which makes the model a general Gaussian mixture. Figure 2 shows the average missclassification rate over 32 repetitions for different $L$. The results are consistent with Theorem 3 showing that as $\delta$ increases, the misclassification rate decreases.

## 4 Proofs

Let us first recall a fact from functional analysis. Consider the space of functions $f : [n] \to \mathcal{X}$ and let us equip $[n]$ with the uniform probability measure $\nu_n$. Then, from the theory of Lebesgue-Bochner spaces, $\|f\|_p := (\int \|f(\omega)\|^p_{\mathcal{X}} \, d\nu_n(\omega))^{1/p}$ defines a proper norm on this function space, turning it into a Banach space $L^p(\nu_n; \mathcal{X})$. In particular, the triangle inequality holds for this norm. Note that $\|f\|_p = (\frac{1}{n} \sum^n_{i=1} \|f(i)\|^p_{\mathcal{X}})^{1/p}$. We will frequently invoke the triangle inequality in $L^p(\nu_n, \mathcal{X})$.

Let $\mu^* := \sum_k \pi_k \delta_{\xi^*_k} = \frac{1}{n} \sum^n_{i=1} \delta_{\xi^*_{z_i}}$ be the empirical measure associated with $\{\xi^*_{z_i}\}$. Recalling definition (1) of the population cost function, we have, for any $\xi \in \mathcal{X}^L$,

$$Q(\xi; \mu^*)^p = \sum_{k=1}^K \pi_k \min_{1 \le \ell \le L} \|\xi^*_k - \xi_\ell\|^p = \frac{1}{n} \sum_{i=1}^n \min_{1 \le \ell \le L} \|\xi^*_{z_i} - \xi_\ell\|^p. \tag{9}$$

We start with three lemmas that are proved in the Supplementary Material:

**Lemma 1.** *Let ALG(p) be a k-means algorithm satisfying Assumption 1(b') and let $\widehat{\xi}$ be its output for L clusters. Furthermore, assume $(\frac{1}{n}\sum_{i=1}^{n}\|x_i - \xi_{z_i}^*\|^p)^{1/p} \leq \varepsilon$. Then $Q(\widehat{\xi}; \mu^*) \leq (1+\kappa)\varepsilon$.*

**Lemma 2** (Curvature). *For every $\xi \subset \mathcal{X}^L$ and $\xi^* \in \mathcal{X}^K$, with $L \geq K$,*

$$Q(\xi; \mu^*) \geq \pi_{\min}^{1/p}\Big(d_p(\xi, \xi^*) \wedge \frac{\delta}{2}\Big).$$

**Lemma 3.** *Assume that $\max_{1\leq i\leq n}\|x_i - \xi_{z_i}^*\| \leq \eta$ and $d_\infty(\widehat{\xi}, \xi^*) \leq \gamma$. When $L = K$, if $\delta > 2\gamma + 2\eta$, there exists a bijective function $\omega : [K] \to [K]$ satisfying $\omega(\widehat{z}_i) = z_i$, $\forall i \in [n]$. When $L > K$, if $\delta > 2\gamma + 4\eta$, there exists a surjective function $\omega : [L] \to [K]$ satisfying $\omega(\widehat{z}_i) = z_i$, $\forall i \in [n]$.*

*Proof of Theorem 1.* As $(\frac{1}{n}\sum_{i=1}^{n}\|x_i - \xi_{z_i}^*\|^p)^{1/p} \leq \max_{1\leq i\leq n}\|x_i - \xi_{z_i}^*\| \leq \eta$, combining Lemma 1 and 2, we have

$$\Big(d_p(\widehat{\xi}, \xi^*) \wedge \frac{\delta}{2}\Big) \leq \frac{Q(\widehat{\xi}, \mu^*)}{\pi_{\min}^{1/p}} \leq \frac{(1+\kappa)\eta}{\pi_{\min}^{1/p}}.$$

By the condition on $\delta$ in (4), we have $\delta/2 > (1+\kappa)\eta/\pi_{\min}^{1/p}$. Then, $d_\infty(\widehat{\xi}, \xi^*) \leq d_p(\widehat{\xi}, \xi^*) \leq \gamma := (1+\kappa)\eta/\pi_{\min}^{1/p}$, which also makes the assumption in Lemma 3 that $\delta > 2\gamma + 4\eta$ valid. Finally, the result follows from Lemma 3. $\qquad\square$

*Proof of Theorem 2.* The argument is similar to one that has appeared in recent literature [17, 12, 33]. From the proof of Lemma 1 (in the Supplementary Material), we have

$$Q(\widehat{\xi}; \mu^*) \leq \Big(\frac{1}{n}\sum_{i=1}^{n}\|\xi_{z_i}^* - \widehat{\xi}_{\widehat{z}_i}\|^p\Big)^{1/p} \leq (1+\kappa)\varepsilon.$$

By Lemma 2

$$\Big(d_p(\widehat{\xi}, \xi^*) \wedge \frac{\delta}{2}\Big) \leq \frac{Q(\widehat{\xi}, \mu^*)}{\pi_{\min}^{1/p}} \leq \frac{(1+\kappa)\varepsilon}{\pi_{\min}^{1/p}}.$$

By the separation assumption in (5), $\delta/2 > (1+\kappa)\varepsilon/\pi_{\min}^{1/p}$. Hence $d_p(\widehat{\xi}, \xi^*) \leq (1+\kappa)\varepsilon/\pi_{\min}^{1/p}$. Let $\mathcal{C}_k = \{i : z_i = k\}$, $|\mathcal{C}_k| = n_k$, and set $T_k := \{i \in \mathcal{C}_k : \|\xi_{z_i}^* - \widehat{\xi}_{\widehat{z}_i}\| \leq \delta/c\}$. Letting $S_k = \mathcal{C}_k \setminus T_k$, we obtain

$$|S_k|\delta^p/c^p < \sum_{i\in S_k}\|\xi_{z_i}^* - \widehat{\xi}_{\widehat{z}_i}\|^p \leq \sum_{i=1}^{n}\|\xi_{z_i}^* - \widehat{\xi}_{\widehat{z}_i}\|^p \leq n(1+\kappa)^p\varepsilon^p.$$

Therefore,

$$\frac{|S_k|}{n_k} < \frac{n(1+\kappa)^pc^p\varepsilon^p}{n_k\delta^p} \leq 1.$$

The last inequality is by assumption $\delta > (1+\kappa)c\varepsilon/\pi_{\min}^{1/p}$. Hence, $T_k$ is not empty. Furthermore, we argue that if $i \in T_k$ and $j \in T_\ell$ for $k \neq \ell$, i.e. $z_i \neq z_j$, then $\widehat{z}_i \neq \widehat{z}_j$. Assume otherwise, that is, $\widehat{z}_i = \widehat{z}_j$. Then

$$\|\xi_k^* - \xi_\ell^*\| \ \leq \ \|\xi_k^* - \widehat{\xi}_{\widehat{z}_i}\| + \|\xi_\ell^* - \widehat{\xi}_{\widehat{z}_j}\| \ \leq \ 2\delta/c \ < \ \delta$$

causing a contradiction.

Let $\mathcal{L}_k := \{\widehat{z}_i : i \in T_k\}$ and $\mathcal{L} = \bigcup_{k=1}^{K}\mathcal{L}_k$. Define a function $\omega : \mathcal{L} \to [K]$ by setting $\omega(\ell) = k$ for all $\ell \in \mathcal{L}_k$ and $k \in [K]$. By the property of $\{T_k\}$ shown above, $\mathcal{L}_k, k \in [K]$ are disjoint and nonempty sets. This implies that $\omega$ is well-defined and surjective. Extend $\omega$ to a surjective $\omega : [L] \to [K]$ by arbitrarily defining it for $[L] \setminus \mathcal{L}$. Note that $\widehat{z}_i \in \mathcal{L}_k$ implies $z_i = k$. It follows that $\omega(\widehat{z}_i) = z_i$ for all $\widehat{z}_i \in \mathcal{L}$, and

$$\frac{1}{n}\sum_{i=1}^{n}\mathbb{1}\{z_i \neq \omega(\widehat{z}_i)\} \leq \frac{n - |\mathcal{L}|}{n} = \sum_{k=1}^{K}\frac{|S_k|}{n} < \frac{K(1+\kappa)^pc^p\varepsilon^p}{\delta^p}.$$

The result follows. $\qquad\square$

*Proof of Theorem 3.* By assumption, $\kappa$-approximation holds for both $K$ and $L$ clusters. Then,

$$\widehat{Q}(\widehat{\xi}) \leq \kappa \, \widehat{Q}_{\min}^{(L)}, \quad \text{where} \quad \widehat{Q}_{\min}^{(L)} := \min_{\xi \in \mathcal{X}^L} \widehat{Q}(\xi).$$

Since $\widehat{Q}_{\min}^{(L)} \leq (\frac{1}{n} \sum_{i=1}^{n} \|x_i - \widetilde{\xi}_{\widetilde{z}_i}\|^p)^{1/p} \leq \varepsilon$, by the triangle inequality in $L^p(\nu_n, \mathcal{X})$,

$$\left( \frac{1}{n} \sum_{i=1}^{n} \|\widetilde{\xi}_{\widetilde{z}_i} - \widehat{\xi}_{\widehat{z}_i}\|^p \right)^{1/p} \leq \left( \frac{1}{n} \sum_{i=1}^{n} \|x_i - \widetilde{\xi}_{\widetilde{z}_i}\|^p \right)^{1/p} + \left( \frac{1}{n} \sum_{i=1}^{n} \|x_i - \widehat{\xi}_{\widehat{z}_i}\|^p \right)^{1/p} \leq (1 + \kappa)\varepsilon.$$

Let $T_k := \{ i \in \mathcal{C}_k : \|\widetilde{\xi}_{\widetilde{z}_i} - \widehat{\xi}_{\widehat{z}_i}\| \leq \delta/c \}$ and $S_k = \mathcal{C}_k \setminus T_k$. Then,

$$|S_k| \delta^p / c^p < \sum_{i \in S_k} \|\widetilde{\xi}_{\widetilde{z}_i} - \widehat{\xi}_{\widehat{z}_i}\|^p \leq \sum_{i=1}^{n} \|\widetilde{\xi}_{\widetilde{z}_i} - \widehat{\xi}_{\widehat{z}_i}\|^p \leq n(1+\kappa)^p \varepsilon^p.$$

Therefore,

$$\frac{|S_k|}{n_k} < \frac{n(1+\kappa)^p c^p \varepsilon^p}{n_k \delta^p} \leq 1$$

The last inequality is by assumption $\delta \geq (1+\kappa)c\varepsilon / \pi_{\min}^{1/p}$. Hence $T_k$ is not empty. Next we argue that if $i \in T_k$, $j \in T_\ell$ for $k \neq \ell$, i.e. $z_i \neq z_j$, then $\widehat{z}_i \neq \widehat{z}_j$. Assume otherwise, that is $\widehat{z}_i = \widehat{z}_j$. Since $\widetilde{z}$ is a refinement of $z$, $z_i \neq z_j$ implies $\widetilde{z}_i \neq \widetilde{z}_j$ and $\widetilde{\omega}(\widetilde{z}_i) \neq \widetilde{\omega}(\widetilde{z}_j)$. By the triangle inequality,

$$\|\widetilde{\xi}_{\widetilde{z}_i} - \widetilde{\xi}_{\widetilde{z}_j}\| \leq \|\widetilde{\xi}_{\widetilde{z}_i} - \widehat{\xi}_{\widehat{z}_i}\| + \|\widetilde{\xi}_{\widetilde{z}_j} - \widehat{\xi}_{\widehat{z}_j}\| \leq 2\delta/c < \delta$$

causing a contradiction. The rest of the proof follows that of Theorem 2. $\qquad\square$

*Proof of Proposition 2.* Let $m_k$ be the mean of $\mathbb{Q}_k$. Then, $\mathbb{P}(|t_i - m_{z_i}| > t) \leq 2e^{-t^2/2\sigma^2}$. Let $M = \sqrt{6\sigma^2 \log n}$. By union bound, with probability $\geq 1 - 2n^{-2}$ we have $|t_i - m_{z_i}| \leq M$ for all $i \in [n]$. We can cover the set $[-M, M] \subset \mathbb{R}$, with $L' = M/\varepsilon$ 1-D balls of radius $\varepsilon$. (Without loss of generality, we assume that $L'$ is an integer for simplicity.) Let $\mathcal{T} = \{\tau_1, \ldots, \tau_{L'}\}$ one such cover and note that $m_k + \mathcal{T}$ is an $\varepsilon$-cover of $m_k + [-M, M]$. Let $\pi_k : \mathbb{R} \to (m_k + \mathcal{T})$ be the projection from $\mathbb{R}$ onto $m_k + \mathcal{T}$. Then, $\|\gamma_{z_i}(t_i) - \gamma_{z_i}(\pi_{z_i}(t_i))\| \leq \rho|t_i - \pi_{z_i}(t_i)| \leq \rho\varepsilon$, assuming that $\varepsilon \leq 1/\rho$.

Let $z_i' := \operatorname{argmin}_{\ell' \in [L']} |t_i - (m_{z_i} + \tau_{\ell'})|$ so that $\pi_{z_i}(t_i) = m_{z_i} + \tau_{z_i'}$. Then let $L_n = KL'$ and fix a bijection $\phi : [L_n] \to [K] \times [L']$ and define the labels $\widetilde{z}_i = \phi^{-1}(z_i, z_i')$. Also consider the map $\omega_0 : [K] \times [L'] \to [K]$ given by $\omega_0(k, \ell') = k$ and set $\widetilde{\omega} := \omega_0 \circ \phi$ which is a surjective map from $[L_n]$ to $[K]$ satisfying $\widetilde{\omega}(\widetilde{z}_i) = z_i$. For $\ell \in [L_n]$ with $\phi(\ell) = (k, \ell')$, define $\widetilde{\xi}_\ell = \gamma_k(m_k + \tau_{\ell'})$. Then, we have $\widetilde{\xi}_{\widetilde{z}_i} = \gamma_{z_i}(m_{z_i} + \tau_{z_i'}) = \gamma_{z_i}(\pi_{z_i}(t_i))$, hence the above argument gives $\|\gamma(t_i) - \widetilde{\xi}_{\widetilde{z}_i}\| \leq \rho\varepsilon$. It is also clear that the the separation condition (7) is satisfied since by construction if $\widetilde{\omega}(\ell_1) \neq \widetilde{\omega}(\ell_2)$ with $\phi(\ell_1) = (k_1, \ell_1')$ and $\phi(\ell_2) = (k_2, \ell_2')$, then $k_1 \neq k_2$ hence $\widetilde{\xi}_{\ell_1}$ and $\widetilde{\xi}_{\ell_2}$ lie on different manifolds (on $\mathcal{C}_{k_1}$ and $\mathcal{C}_{k_2}$). It follows that conclusion (8) of Theorem 3 holds for $p = 2$ and, say, $c = 3$ but with $\varepsilon$ replaced with $\rho\varepsilon$. Take $\varepsilon = (c_1\sqrt{n})^{-1}$ for constant $c_1$ to be determined. Let $c_2 = 3\rho(1+\kappa)/\delta$. As long as $n\pi_{\min} > (c_2/c_1)^2$, the separation condition in (8) is satisfied and we have $\operatorname{Miss}(z, \widehat{z}) \leq K(c_2/c_1)^2/n$. Hence, as long as $c_1 > \sqrt{K}c_2$, we will have $\operatorname{Miss}(z, \widehat{z}) < 1/n$ which implies $\operatorname{Miss}(z, \widehat{z}) = 0$. We also need to satisfy $\varepsilon < 1/\rho$ that is $c_1 \geq \rho/\sqrt{n}$. Taking $c_1 = \sqrt{K}c_2 + \rho$ satisfies all the required constraints on $c_1$. The required number of clusters is

$$L_n = KL' = KM/\varepsilon \leq 3K\sigma c_1 \sqrt{n \log n},$$

which proves the result with $C = 3K\sigma c_1$. Note that since $c_2/c_1 < 1$ and $n\pi_{\min} \geq 1$, the condition $n\pi_{\min} > (c_2/c_1)^2$ is automatically satisfied. The proof is complete. $\qquad\square$

## Acknowledgments and Disclosure of Funding

This work was supported by the NSF CAREER grant DMS-1945667. We thank the referees for their helpful comments that lead to the improvement of the paper.

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
