# Appendix for "Label consistency in overfitted generalized $k$-means"

**Linfan Zhang**
Department of Statistics
University of California, Los Angeles
Los Angeles, CA 90095
linfanz@g.ucla.edu

**Arash A. Amini**
Department of Statistics
University of California, Los Angeles
Los Angeles, CA 90095
aaamini@ucla.edu

This appendix contains further discussion of the results, the remaining proofs, details of some examples and additional numerical experiments.

## A Discussion

Proposition 2 and 3 show that perfect refinement for sub-gaussian mixture-of-curves model can be achieved when the number of clusters grows as $L_n = O(\sqrt{n \log n})$. To the best of our knowledge, this is the first such result in the literature, that is, an upper bound on the minimum number of clusters needed to achieve a perfect refinement of the true clusters. What remains for future investigations to determine is how tight this bound is. Empirically, we have found examples of the mixture-of-curves model for which $L_n \asymp 1$ seems to enough, but also an example where $L_n \asymp \sqrt{n \log n}$ seems to be the required scaling. Figure S1(a) shows a noisy circle-torus model (cf. Section D.1) with $R = 10, r = 2$ and $\sigma = 1$ that demonstrates the scaling $L_n \asymp \sqrt{n \log n}$. Here, we plot the average misclassification rate over 32 repetitions vs $L_n/\sqrt{n \log n}$ for various $n$. The fact that these plots coincide with each other for different $n$ suggests that there is sharp threshold $\tau_n = C_1 \sqrt{n \log n}$ such that with $L_n > \tau_n$, perfect refinement recovery is possible and with $L_n < \tau_n$, impossible. Figure S1(b) shows an example that exhibits $L_n \asymp 1$ threshold: A line-circle model with parameters $\delta = 4$, $\sigma = 1$ and line standard deviation = 7.

The fact that, empirically, there are examples for which $L_n$ has to grow as fast as $\sqrt{n \log n}$ for a perfect refinement recovery, suggests that the result of Proposition 2 may be sharp up to constants, over the class of mixture-of-curves distributions considered.

## B Connection to distribution stability

The distribution stability for the $K$-means assumes the following [1]:

$$\|x - \xi_k^*\|^2 \geq \beta \cdot \frac{\text{OPT}_K}{n_k}, \quad \text{for all } x \notin C_k,$$

where $\text{OPT}_K = \sum_{i=1}^n \|x_i - \xi_{z_i}^*\|^2$ for the $K$-means optimal cluster labels $\{z_i\} \subset [K]^n$ and optimal centers $\{\xi_k^*\}$. Here, $C_k = \{i : z_i = k\}$ and $n_k = |C_k|$.

In our setting, we do not necessarily need to work with the optimal $K$-means clustering. So let us generalize the notion as follows: The data $\{x_i\}$ is $\beta$-distributed with respect to cluster labels $\{z_i\}$ and centers $\{\xi_k^*\}$ if

$$\|x - \xi_k^*\|^2 \geq \beta \cdot \sum_{i=1}^n \|x_i - \xi_{z_i}^*\|^2/n_k, \quad \text{for all } x \notin C_k,$$

35th Conference on Neural Information Processing Systems (NeurIPS 2021), Sydney, Australia.

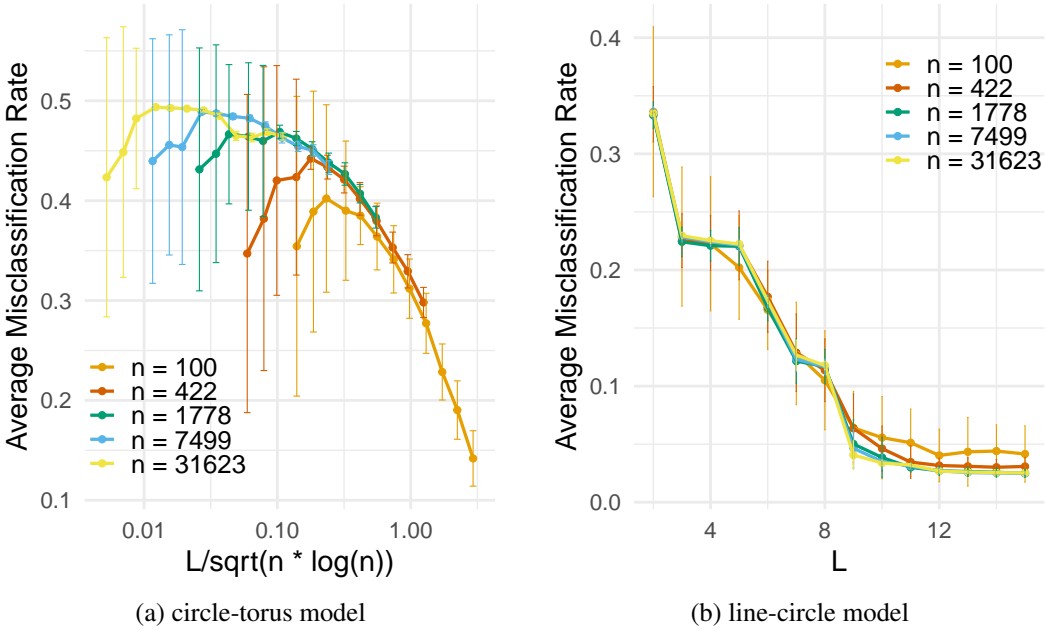

(a) circle-torus model        (b) line-circle model

Figure S1: Examples of mixture-of-curve models that exhibit (a) $L_n \asymp \sqrt{n \log n}$ and (b) $L_n = O(1)$ refinement recovery threshold.

where $C_k = \{i : z_i = k\}$ and $n_k = |C_k|$. Setting $\frac{1}{n} \sum_{i=1}^{n} \|x_i - \xi_{z_i}^*\|^2 = \varepsilon^2$ and recalling $\pi_k = n_k/n$, the condition is equivalent to

$$\|x - \xi_k^*\| \geq \frac{\sqrt{\beta} \cdot \varepsilon}{\sqrt{\pi_k}}, \quad \text{for all } x \notin C_k. \tag{S1}$$

Let us strengthen the condition slightly and consider the following notion instead

$$\|x - \xi_k^*\| \geq \frac{\sqrt{\beta} \cdot \varepsilon}{\sqrt{\pi_{\min}}}, \quad \text{for all } x \notin C_k, \tag{S2}$$

where $\pi_{\min} = \min_k \pi_k$. (This is without loss of generality: We could have stated our results with separate center separation parameters for each cluster, i.e., $\delta_k = \min_{\ell \neq k} \|\xi_k^* - \xi_\ell^*\|$, in which case we could directly compare with the original version (S1). We opted for the simpler global center separation in the paper for simplicity.)

Now assume that the data is $\beta$-distributed and in addition:

(D1) For all distinct pairs $(k, \ell)$, there is $x \in C_\ell$ such that $\|x - \xi_k^*\| \leq \|\xi_\ell^* - \xi_k^*\|$.

That is, every cluster $C_\ell$ has points which are closer than $\xi_\ell^*$ to the centers of other clusters. Then, it follows that

$$\frac{\delta}{\varepsilon} \geq \frac{\sqrt{\beta}}{\sqrt{\pi_{\min}}} \tag{S3}$$

which is our separation condition. (Recall that $\delta = \min_{k \neq \ell} \|\xi_k^* - \xi_\ell^*\|$).

In fact, in the presence of (D1), we can relax $\beta$-distribution stability as follows: Assume (D1) and for the $x$ in (D1) assume that the inequality in (S2) holds. Then, our separation condition (S3) follows. Note that (D1) is quite mild and one expects it to hold almost always if there is some full-dimensional randomness in the distribution of the points in a cluster.

Alternatively, our separation condition can be written equivalently as

$$\|x - \xi_k^*\| \geq \frac{\sqrt{\beta} \cdot \varepsilon}{\sqrt{\pi_{\min}}}, \quad \text{for all } x \in \{\xi_\ell^*\}_{\ell \neq k} \tag{S4}$$

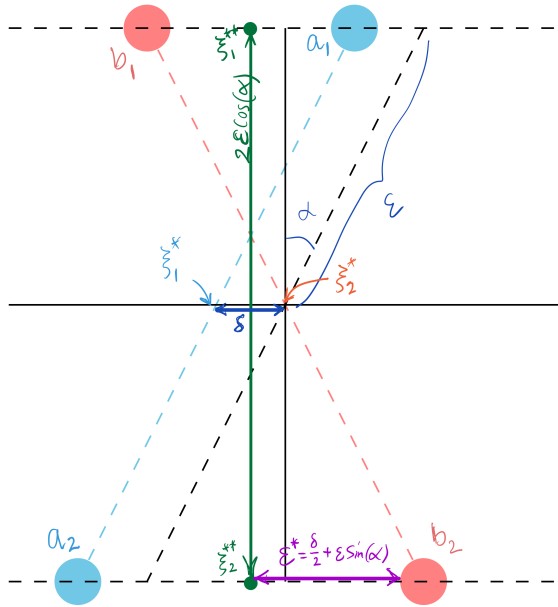

Figure S2: The geometry of the dataset in Proposition 1

Comparing (S4) and (S2), the conditions are somewhat close, but different. Neither condition directly follow from the other one in general.

Note also that although in the discussion above, we refer to $\xi_k^*$ as the center of $C_k$, in our general setting $\xi_k^*$ need not be the optimal center $\frac{1}{n_k} \sum_{i \in C_k} x_i$.

## C   Proof of Proposition 1

For $\alpha \in [0, \pi/2)$, consider a constellation of points in $\mathbb{R}^2$ at locations $a_1 = (\varepsilon \sin \alpha - \delta, \varepsilon \cos \alpha)$, $a_2 = (-\varepsilon \sin \alpha - \delta, -\varepsilon \cos \alpha)$, $b_1 = (-\varepsilon \sin \alpha, \varepsilon \cos \alpha)$ and $b_2 = (\varepsilon \sin \alpha, -\varepsilon \cos \alpha)$. Assume that $n/4$ of the data points are on each of the points $a_1, a_2, b_1$ and $b_2$. Assume that data points in $\{a_1, a_2\}$ form cluster 1 and points in $\{b_1, b_2\}$ form cluster 2. That is, this is the true cluster labels as specified by an external source. The true cluster centers are then at locations $\xi_1^* = (-\delta, 0)$ and $\xi_2^* = (0, 0)$. We also have $(\frac{1}{n} \sum_i \|x_i - \xi_{z_i}^*\|^2)^{1/2} = \varepsilon$ for true cluster labels $\{z_i\}$. Now take $\delta = \varepsilon \sin \alpha$. Figure S2 shows the geometry of this construction.

To show the result, it is enough to use Theorem 2 with properly chosen (fake) centers on the above dataset. In particular, we are going to show that a 2-factor $k$-means algorithm has a small misclassification rate with respect to a new clustering that puts points $\{a_1, b_1\}$ in one cluster and $\{a_2, b_2\}$ in another cluster. Consider "fake" centers $\xi_1^{**} = (a_1 + b_1)/2$ and $\xi_2^{**} = (a_2 + b_2)/2$. Then, the new separation is $\delta^* = 2\varepsilon \cos \alpha$ and the new deviation can be taken to be $\varepsilon^* = \delta/2 + \varepsilon \sin \alpha = (3/2)\varepsilon \sin \alpha$ guaranteeing that $(\frac{1}{n} \sum_i \|x_i - \xi_{y_i}^{**}\|^2)^{1/2} \leq \varepsilon^*$ where $\{y_i\}$ are labels relative to the new clustering.

Applying Theorem 2 with $\kappa = p = 2$, $c = 2.1$ and $\pi_{\min} = 1/2$, as long as $\delta^*/\varepsilon^* \geq 9 > 3\sqrt{2}c$, the misclassification rate to the new clustering is bounded above as Miss$^* \leq 80(\varepsilon^*/\delta^*)^2$. We have $\varepsilon^*/\delta^* = (3/4) \tan \alpha$. Thus, for $\alpha \leq \tan^{-1}(4/27)$ we have Miss$^* \leq 45(\tan \alpha)^2$ w.r.t. to clustering $\{\{a_1, b_1\}, \{a_2, b_2\}\}$. Hence, w.r.t. the original clustering, $\frac{1}{2} \geq$ Miss $\geq \frac{1}{2} - 45(\tan \alpha)^2$, which can be made arbitrarily close to $\frac{1}{2}$ by choosing $\alpha$ small enough.

To see the last step above, let $q_1, q_2, q_3, q_4$ be the fractions of misclassified nodes from each of the four categories $a_1, a_2, b_1, b_2$, w.r.t. to the new clustering (i.e., $\{y_i\}$). The above argument shows that $\frac{1}{4}(q_1 + q_2 + q_3 + q_4) \leq 45(\tan \alpha)^2$. The misclassification rate to the original clustering (i.e., $\{z_i\}$) is then

$$\text{Miss} = \frac{1}{n} \left( \frac{n}{4}(1 - q_{i_1}) + \frac{n}{4}(1 - q_{i_2}) \right) = \frac{1}{2} - \frac{1}{4}(q_{i_1} + q_{i_2}) \geq \frac{1}{2} - 45(\tan \alpha)^2$$

where $\{i_1, i_2\}$ is a pair of distinct elements from $\{1, 2, 3, 4\}$. This proves the lower bound. The upper bound Miss $\leq 1/2$ always holds due to the minimization over permutations in the definition of the misclassification rate.

Since for $\beta \in (0, 1/2)$, $\sqrt{\beta/45} \leq 4/27$, we only need $\alpha \leq \tan^{-1}(\sqrt{\beta/45})$ to have $\frac{1}{2} \geq$ Miss $\geq \frac{1}{2} - \beta$. Recalling that $\delta/\varepsilon = \sin\alpha$, this shows that one can take $c_2(\beta) = \sin(\tan^{-1}(\sqrt{\beta/45}))$ in the statement of the lower bound.

For the claim regarding perfect recovery with $L = 4$ clusters, take $\xi_1^{**} = a_1$, $\xi_2^{**} = b_1$, $\xi_3^{**} = a_2$ and $\xi_4^{**} = b_2$ and apply Theorem 1, noting that $\delta^* = \min_{i \neq j} \|\xi_i^{**} - \xi_j^{**}\| > 0$ while we can take $\varepsilon^* = 0$.

# D   Experiment details

The code for numerical experiments are executed in R [3] version 4.0.3 on a Linux system with 48 CPU cores. The code is provided as a ZIP file as part of the supplementary material. We use the `kmeans` function in base R and go with the default algorithm of Hartigan and Wong [2]. We set the number of random starts to `nstart = 20` and the maximum number of iterations allowed to `iter.max = 200`.

## D.1   Circle-torus model

The circle-torus model is a mixture of two parts: (1) The uniform distribution on the circumference of a circle in the $xy$-plane, at the origin, and (2) a torus built around this circle. Parametrically, these two clusters can be defined via the following equations,

$$
\begin{aligned}
x_1 &= R\cos(t) & & & x_2 &= (R + r\cos(mt))\cos(t) \\
y_1 &= R\sin(t) & & \text{and} & y_2 &= (R + r\cos(mt))\sin(t) & & \text{(S5)}\\
z_1 &= 0 & & & z_2 &= r\sin(mt).
\end{aligned}
$$

Here $R$ is the radius of the circle on the plane and also the distance from the center of the tube to the center of the torus. $r$ is the radius of the tube and it is also the minimal distance between two clusters. We also created a noisy version by adding $N(0, \sigma^2 I_3)$ to the model. Figure 3 shows the geometry of the two clusters in the case $R = 3, r = 1$ and $\sigma = 0$. The other two panels in Figure 3 show the average missclassification rate over 32 repetitions versus $\delta := r$, for both the noiseless and noisy ($\sigma = 1$) circle-torus model. In both cases, we let $R = 3$ and very $r$ (i.e., $\delta$), from 0.1 to 10. In Figure S3, we include additional scatter plots of the circle-torus model for various settings of the parameters $(R, r, \sigma)$. Figure S3(a) is the noisy version of Figure 3(a) with noise level $\sigma = 0.1$. Figure S3(b) shows that for sufficiently small $r$ and high noise, the two clusters are nearly indistinguishable. Figure S3(c) shows the scatter plot for $R = 3$ and $r = 10$; it is an example of how the model looks like when $R < r$.

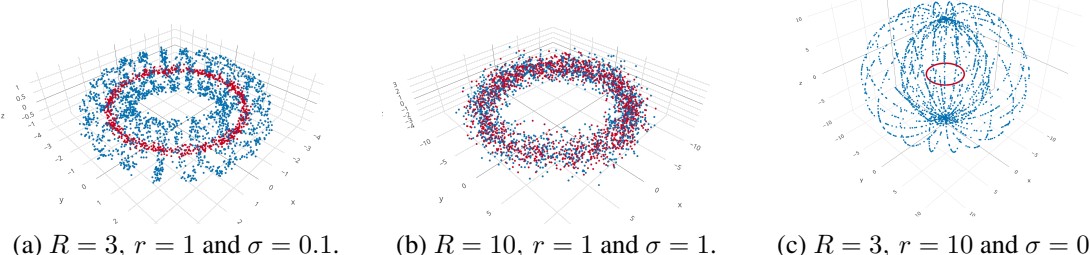

(a) $R = 3$, $r = 1$ and $\sigma = 0.1$.     (b) $R = 10$, $r = 1$ and $\sigma = 1$.     (c) $R = 3$, $r = 10$ and $\sigma = 0$.

Figure S3: Scatter plots for the circle-torus model. True clusters are distinguished by their color.

# E   Norm of a sub-gaussian vector

We first recall the definition of a sub-gaussian vector: A random vector $X = (X_1, \ldots, X_d) \in \mathbb{R}^d$ is sub-gaussian if the one-dimensional marginals $u^T X$ are sub-gaussian random variables for all $u \in$

$\mathbb{R}^d$ [4, Definition 3.4.1]. The sub-gaussian norm of $X$ is defined as $\|X\|_{\psi_2} = \sup_{u \in S^{d-1}} \|u^T X\|_{\psi_2}$, where $\|\cdot\|_{\psi_2}$ denotes the sub-gaussian norm of a random variable and $S^{d-1}$ the unit sphere in $\mathbb{R}^d$. Alternatively, we can define a sub-gaussian vector with parameter $\sigma$, as a random vector satisfying $\mathbb{P}(|u^T X| \geq t) \leq 2\exp(-\frac{t^2}{2\sigma^2})$ for all $u \in S^{d-1}$ and $t \geq 0$. We will have $\sigma \asymp \|X\|_{\psi_2}$. We also use $\|\cdot\|_{\psi_1}$ for the sub-exponential norm of a random variable. For any random variable, we have $\|Y^2\|_{\psi_1} = \|Y\|_{\psi_2}^2$ [4, Lemma 2.7.6]. Below we apply this fact with $Y = \|X\| = (\sum_{i=1}^d X_i^2)^{1/2}$, leading to the following useful lemma.

**Lemma S1.** *Assume that $X \in \mathbb{R}^d$ is a sub-gaussian random vector with parameter $\sigma$. Then, $\|X\|$ is sub-gaussian with parameter $\lesssim \sigma\sqrt{d}$. In fact, for some universal constant $C > 0$,*
$$\|\|X\|\|_{\psi_2} \leq C\sigma\sqrt{d}, \quad \|\|X\|^2\|_{\psi_1} \leq C^2\sigma^2 d.$$

*Proof.* We have $\|\|X\|^2\|_{\psi_1} \leq \sum_{i=1}^d \|X_i^2\|_{\psi_1} = \sum_{i=1}^d \|X_i\|_{\psi_2}^2 \leq dC^2\sigma^2$, for some universal constant $C^2 > 0$. The first inequality is the triangle inequality for $\|\cdot\|_{\psi_1}$ and the second by the equivalence of the sub-gaussian norm and sub-gaussian parameter. Next, we note that $\|\|X\|\|_{\psi_2} = \sqrt{\|\|X\|^2\|_{\psi_1}}$ and the result follows. $\qquad\square$

## F  Details of the sub-gaussian mixture example

By assumption, $w_i$ is a sub-gaussian vector with parameter $\sigma_i$. Then, by Lemma S1, $\|w_i\|^2/d$ is sub-exponential with sub-exponential norm $\lesssim \sigma_i^2$. By the Bernstein inequality for sub-exponential variables [4, Corollary 2.8.3],
$$\mathbb{P}\Big(\frac{1}{n}\Big(\sum_{i=1}^n \frac{\|w_i\|^2}{d} - \alpha_i^2\Big) > t\Big) \leq \exp\Big(-cn\min\Big(\frac{t^2}{\sigma_{\max}^4}, \frac{t}{\sigma_{\max}^2}\Big)\Big).$$
Let $t = \bar{\alpha}_n$, and recall that $\bar{\alpha}_n^2/\sigma_{\max}^2 \leq C$. Then, for a constant $c_1 > 0$,
$$\mathbb{P}\Big(\frac{1}{n}\sum_{i=1}^n \frac{\|w_i\|^2}{d} > 2\bar{\alpha}_n^2\Big) \leq \exp\Big(-c_1 n \frac{\bar{\alpha}_n^2}{\sigma_{\max}^4}\Big).$$
In the Gaussian case $w_i \sim N(0, \Sigma_i)$, it is not hard to see that $w_i$ is a sub-guassian vector with parameter $\|\Sigma_i\|_{op}$. Therefore, in Gaussian mixtures, we have $\sigma_{\max} = \max_i \|\Sigma_i\|_{op}$ and $\bar{\alpha}_n^2 = \sum_{i=1}^n \frac{1}{n}\operatorname{tr}(\Sigma_i)/d$.

## G  Extension of Proposition 3

For random vector $x$ with distribution $\mu_{\mathcal{C}}$ on some subset $\mathcal{C} \subset \mathbb{R}^d$, let $N_{\mu_{\mathcal{C}}}(\varepsilon)$ be the smallest integer for which there is a high probability $\varepsilon$-cover of $x$, that is, a finite subset $\mathcal{N} \subset \mathcal{C}$ such that $\mathbb{P}(\min_{y \in \mathcal{N}} \|x - y\| \leq \varepsilon) \geq 1 - n^{-2}$. We refer to $N_{\mu_{\mathcal{C}}}(\varepsilon)$ as the stochastic covering number of $\mu_{\mathcal{C}}$. We have the following extension of Proposition 3.

**Proposition S1.** *Assume that $\{x_i\}_{i=1}^n$ are independent draws from a $K$-mixture where the $k$th component is a distribution $\mu_{\mathcal{C}_k}$ on a subset $\mathcal{C}_k \subset \mathbb{R}^{r_k}$. Let $z_i$ be the label of $x_i$ so that $x_i \mid z_i = k \sim \mu_{\mathcal{C}_k}$. Assume that*
$$\min_{x \in \mathcal{C}_k, \, y \in \mathcal{C}_{k'}} \|x - y\| \geq \delta > 0, \quad \text{for all } k \neq k'.$$
*Let $N_{\mu_{\mathcal{C}_k}}(\varepsilon)$ be the stochastic covering number of $\mu_{\mathcal{C}_k}$. Then, there exist a constant $C = C(K, \delta, \kappa)$ such that any ALG(p), satisfying Assumption 1, applied with $L_n = \sum_{k=1}^K N_{\mu_{\mathcal{C}_k}}(Cn^{-1/p})$ clusters, recovers a perfect refinement of $z = (z_i)$ with probability $\geq 1 - n^{-1}$.*

*Proof.* Let $\mathcal{N}_k \subset \mathcal{C}_k$ be the $\varepsilon$-net that realizes the stochastic $\varepsilon$-covering number of $\mu_{\mathcal{C}_k}$ and let $\pi_k : \mathcal{C}_k \to \mathcal{N}_k$ be the corresponding projection operator. Then, for any $i \in [n]$ for which $z_i = k$, we have $\mathbb{P}(\|x_i - \pi_k(x_i)\| > \varepsilon) \leq n^{-2}$. By union bound, we have $\|x_i - \pi_{z_i}(x_i)\| \leq \varepsilon$ for all $i \in [n]$ with probability at least $1 - n^{-1}$. The collection of the fake centers $\{\widetilde{\xi}_\ell\}_{\ell=1}^{L_n}$ can be taken to be the union of the nets $\bigcup_{k=1}^K \mathcal{N}_k$ with cardinality $L_n = \sum_k N_{\mu_{\mathcal{C}_k}}(\varepsilon)$. The rest of the proof follows those of Propositions 2 and 3 with $\varepsilon = (c_1 n^{1/p})^{-1}$, $c_2 = 3(1+\kappa)/\delta$ and $c_1 = K^{1/p}c_2$. (Note that there is no condition $\varepsilon < 1/\rho$ that needs to be satisfied in this case.) $\qquad\square$

## H   Remaining proofs

*Proof of Corollary 1.* We first construct fake centers $(\widetilde{\xi}_\ell)$ for $(x_i)$ as in the proof of Proposition 2 and treat them as the fake centers for $y_i$. By the triangle inequality,

$$\Big(\frac{1}{n}\sum_{i=1}^n \|y_i - \widetilde{\xi}_{\widetilde{z}_i}\|^2\Big)^{1/2} \leq \Big(\frac{1}{n}\sum_{i=1}^n \|x_i - \widetilde{\xi}_{\widetilde{z}_i}\|^2\Big)^{1/2} + \Big(\frac{1}{n}\sum_{i=1}^n \|w_i/\sqrt{d}\|^2\Big)^{1/2} \leq \varepsilon + \sqrt{2}\bar{\alpha}_n$$

holds with probability at least $1 - p_n - n^{-1}$. The result follows by applying Theorem 3. $\qquad\square$

*Proof of Proposition 3.* The proof follows that of Proposition 2. We only highlight the differences. When $z_i = k$, by Lemma S1, $\|t_i - m_k\|$ is sub-gaussian with parameter $\leq c_0\sigma\sqrt{r_k}$ for some universal constant $c_0 > 0$. Thus, we have $\mathbb{P}(\|t_i - m_k\| \geq t) \leq 2e^{-c_0 t^2/(r_k\sigma^2)}$. Let $M = \sqrt{3c_0 r\sigma^2 \log n}$. By union bound, with probability at least $1 - 2n^{-2}$, we have $\|t_i - m_{z_i}\| \leq M$ for all $i \in [n]$. The $\varepsilon$-cover has to be constructed for $\{u : \|u\| \leq M\}$ in the $\ell_2$ norm, which can be done with a net of size at most $L' = (1 + 2M/\varepsilon)^r$. Take $\varepsilon = (c_1 n^{1/p})^{-1}$ and let $c_2 = 3\rho(1+\kappa)/\delta$. As long as $n\pi_{\min} > (c_2/c_1)^p$, the separation condition in (8) is satisfied and we have $\mathrm{Miss}(z, \widehat{z}) \leq K(c_2/c_1)^p/n$. Hence, as long as $c_1 > K^{1/p}c_2$, we will have $\mathrm{Miss}(z, \widehat{z}) < 1/n$ which implies $\mathrm{Miss}(z, \widehat{z}) = 0$. We also need to satisfy $\varepsilon < 1/\rho$ that is $c_1 \geq \rho/\sqrt{n}$. Taking $c_1 = K^{1/p}c_2 + \rho$ satisfies all the required constraints on $c_1$. The required number of clusters is

$$L_n = KL' = K(1 + 2M/\varepsilon)^r = K(1 + 2c_1\sqrt{3c_0 r\sigma^2}n^{1/p}\sqrt{\log n})^r$$
$$\leq C(n^{1/p}\sqrt{\log n})^r$$

for $C = K(2 + 2c_1\sqrt{3c_0 r\sigma^2})^r$. Here, we have used $1 \leq 2n^{1/p}\sqrt{\log n}$ for $n \geq 2$. Note that since $c_2/c_1 < 1$ and $n\pi_{\min} \geq 1$, the condition $n\pi_{\min} > (c_2/c_1)^p$ is automatically satisfied. The proof is complete. $\qquad\square$

## I   Proofs of the lemmas

*Proof of Lemma 1.* Recall that $\widehat{\xi}$ is the output of ALG for $L$ clusters. Let $\widehat{\xi}^{(K)}$ be the output of the ALG for $K$ clusters. Then, since $L \geq K$,

$$\widehat{Q}(\widehat{\xi}) \leq \widehat{Q}(\widehat{\xi}^{(K)}) \leq \kappa \widehat{Q}_{\min}^{(K)}, \quad \text{where} \quad \widehat{Q}_{\min}^{(K)} := \min_{\xi \in \mathcal{X}^K} \widehat{Q}(\xi).$$

The first inequality is by the monotonicity of ALG and the second by its constant-factor approximation property. Since by assumption $\xi^* \in \mathcal{X}^K$, we have

$$\widehat{Q}_{\min}^{(K)} \leq \widehat{Q}(\xi^*) \leq \Big(\frac{1}{n}\sum_{i=1}^n \|x_i - \xi_{z_i}^*\|^p\Big)^{1/p} \leq \varepsilon.$$

It follows that $\widehat{Q}(\widehat{\xi}) \leq \kappa\varepsilon$. Recalling (9) and noting that $\widehat{Q}(\widehat{\xi}) = \big(\frac{1}{n}\sum_{i=1}^n \|x_i - \widehat{\xi}_{\widehat{z}_i}\|^p\big)^{1/p}$, we have

$$Q(\widehat{\xi}; \mu^*) = \Big(\frac{1}{n}\sum_{i=1}^n \min_{\ell \in [L]} \|\xi_{z_i}^* - \widehat{\xi}_\ell\|^p\Big)^{1/p} \leq \Big(\frac{1}{n}\sum_{i=1}^n \|\xi_{z_i}^* - \widehat{\xi}_{\widehat{z}_i}\|^p\Big)^{1/p}$$
$$\leq \Big(\frac{1}{n}\sum_{i=1}^n \|x_i - \xi_{z_i}^*\|^p\Big)^{1/p} + \Big(\frac{1}{n}\sum_{i=1}^n \|x_i - \widehat{\xi}_{\widehat{z}_i}\|^p\Big)^{1/p}$$
$$\leq \varepsilon + \kappa\varepsilon \tag{S6}$$

where the second line is the triangle inequality in the aforementioned $L^p(\nu_n, \mathcal{X})$ space. The proof is complete. $\qquad\square$

*Proof of Lemma 2.* Consider the partition of the space by the Voronoi cells of $\xi = (\xi_\ell)$. Assume first that there is a Voronoi cell that contains at least two distinct elements of $\xi^*$, e.g., $\xi_{k_1}^*$ and $\xi_{k_2}^*$, with $k_1 \neq k_2$, both belonging to the Voronoi cell of $\xi_{\ell_0}$. That is, $\min_\ell \|\xi_k^* - \xi_\ell\| = \|\xi_k^* - \xi_{\ell_0}^*\|$ for

$k = k_1, k_2$. As $\|\xi^*_{k_1} - \xi^*_{k_2}\| \leq \|\xi^*_{k_2} - \xi^*_{\ell_0}\| + \|\xi^*_{k_1} - \xi^*_{\ell_0}\|$, at least one of the $k = k_1, k_2$ satisfy $\|\xi^*_k - \xi^*_{\ell_0}\| \geq \|\xi^*_{k_1} - \xi^*_{k_2}\|/2$, and assume this is true for $k = k_1$, we have

$$Q(\xi; \mu^*) \geq \pi^{1/p}_{\min}\|\xi^*_{k_1} - \xi_{\ell_0}\| \geq \frac{\pi^{1/p}_{\min}}{2}\|\xi^*_{k_1} - \xi^*_{k_2}\| \geq \frac{\pi^{1/p}_{\min}}{2}\delta.$$

Otherwise, each Voronoi cell of $\xi$ contains at most one element of $\xi^*$. On the other hand, each element of $\xi^*$ belongs to at least one Voronoi cell of $\xi$, since the union of Voronoi cells is the whole space. It follows that there are $K$ distinct Voronoi cells of $\xi$, each of which contains exactly one element of $\xi^*$. Thus, there is an injective map $\sigma : [K] \to [L]$ such that $\xi^*_k$ belongs to Voronoi cell of $\xi_{\sigma(k)}$, that is, $\min_\ell \|\xi^*_k - \xi_\ell\| = \|\xi^*_k - \xi_{\sigma(k)}\|$. Then,

$$Q(\xi; \mu^*) \geq \pi^{1/p}_{\min}\Big(\sum_{k=1}^K \|\xi^*_k - \xi_{\sigma(k)}\|^p\Big)^{1/p} \geq \pi^{1/p}_{\min} d_p(\xi, \xi^*).$$

The proof is complete. $\qquad\square$

*Proof of Lemma 3.* By assumption, there exists an injective map $\sigma : [K] \to [L]$ such that

$$\max_{k \in K} \|\xi^*_k - \widehat{\xi}_{\sigma(k)}\| \leq \gamma.$$

Then, $\sigma$ is invertible on $\mathrm{Im}(\sigma) := \{\sigma(k) : k \in [K]\}$, with an inverse denoted as $\sigma^{-1}$. We obtain

$$\|\xi^*_{\sigma^{-1}(\ell)} - \widehat{\xi}_\ell\| \leq \gamma, \quad \forall \ell \in \mathrm{Im}(\sigma). \tag{S7}$$

First assume that $\widehat{z}_i \in \mathrm{Im}(\sigma)$. We prove that $\sigma(z_i) = \widehat{z}_i$ by contradiction. Suppose that $\sigma(z_i) \neq \widehat{z}_i$. Then, we show that $\|x_i - \widehat{\xi}_{\sigma(z_i)}\| < \|x_i - \widehat{\xi}_{\widehat{z}_i}\|$ contradicting $\widehat{z}_i = \underset{\ell}{\mathrm{argmin}}\|x_i - \widehat{\xi}_\ell\|^2$. By the triangle inequality

$$\|x_i - \widehat{\xi}_{\sigma(z_i)}\| \leq \|x_i - \xi^*_{z_i}\| + \|\widehat{\xi}_{\sigma(z_i)} - \xi^*_{z_i}\| \leq \eta + \gamma. \tag{S8}$$

Since $\widehat{z}_i \in \mathrm{Im}(\sigma)$ and $\sigma(z_i) \neq \widehat{z}_i$, we have $\sigma^{-1}(\widehat{z}_i) \neq z_i$. By (S7), $\|\widehat{\xi}_{\widehat{z}_i} - \xi^*_{\sigma^{-1}(\widehat{z}_i)}\| \leq \gamma$. Therefore,

$$\begin{aligned}
\|x_i - \widehat{\xi}_{\widehat{z}_i}\| &\geq \|\widehat{\xi}_{\widehat{z}_i} - \xi^*_{z_i}\| - \|x_i - \xi^*_{z_i}\| \\
&\geq \|\xi^*_{z_i} - \xi^*_{\sigma^{-1}(\widehat{z}_i)}\| - \|\widehat{\xi}_{\widehat{z}_i} - \xi^*_{\sigma^{-1}(\widehat{z}_i)}\| - \eta \\
&\geq \delta - \gamma - \eta.
\end{aligned} \tag{S9}$$

Since by assumption $\delta > 2\gamma + 2\eta$, the claimed contradiction follows by combining (S8) and (S9). Hence, we have $\sigma(z_i) = \widehat{z}_i$ when $\widehat{z}_i \in \mathrm{Im}(\sigma)$. Define $\omega(\cdot) = \sigma^{-1}(\cdot)$ on $\mathrm{Im}(\sigma) \subset [L]$. Then, $\omega$ satisfies $\omega(\widehat{z}_i) = z_i$ whenever $\widehat{z}_i \in \mathrm{Im}(\sigma)$. This finishes proof for the case $L = K$.

Next, we define $\omega$ for $\ell_0 \notin \mathrm{Im}(\sigma)$. Since $\widehat{\xi}$ is an efficient solution, there exists at least one $i \in [n]$ such that $\widehat{z}_i = \ell_0$. When there is only one such $i$, we can just let $\omega(\ell_0) = \omega(\widehat{z}_i) = z_i$. When there are at least two data points $x_i$ and $x_j$ such that $\widehat{z}_i = \widehat{z}_j = \ell_0$, we are going to show, by contradiction, that their true cluster labels must be the same, i.e., $z_i = z_j$. Suppose that $z_i \neq z_j$, then we will show that $\|x_i - \widehat{\xi}_{\ell_0}\| > \|x_i - \widehat{\xi}_{\sigma(z_i)}\|$ which contradicts $x_i$ being in the Voronoi cell of $\widehat{\xi}_{\ell_0}$. Inequality (S8) still holds in this case. Furthermore

$$\begin{aligned}
\|x_i - \widehat{\xi}_{\ell_0}\| &\geq \|\widehat{\xi}_{\ell_0} - \xi^*_{z_i}\| - \|x_i - \xi^*_{z_i}\| \\
&\geq \|x_j - \xi^*_{z_i}\| - \|x_j - \widehat{\xi}_{\ell_0}\| - \eta \\
&\geq \|\xi^*_{z_i} - \xi^*_{z_j}\| - \|x_j - \xi^*_{z_j}\| - \|x_j - \widehat{\xi}_{\ell_0}\| - \eta \\
&\geq \delta - 2\eta - \|x_j - \widehat{\xi}_{\ell_0}\|.
\end{aligned} \tag{S10}$$

Since $x_j$ is in the Voronoi cell of $\widehat{\xi}_{\ell_0}$ and $\ell_0 \notin \mathrm{Im}(\sigma)$, we have $\ell_0 \neq \sigma(z_j)$. Therefore,

$$\begin{aligned}
\|x_j - \widehat{\xi}_{\ell_0}\| &\leq \|x_j - \widehat{\xi}_{\sigma(z_j)}\| \\
&\leq \|x_j - \xi^*_{z_j}\| + \|\widehat{\xi}_{\sigma(z_j)} - \xi^*_{z_j}\| \\
&\leq \eta + \gamma.
\end{aligned} \tag{S11}$$

Combining inequalities (S8), (S10) and (S11) and using the assumption $\delta > 2\gamma + 4\eta$, we get

$$\|x_i - \widehat{\xi}_{\ell_0}\| \;\geq\; \delta - 3\eta - \gamma \;>\; \eta + \gamma \;\geq\; \|x_i - \widehat{\xi}_{\sigma(z_i)}\|$$

which is the claimed contradiction. Therefore, we can define $\omega$ on $[L] \setminus \mathrm{Im}(\sigma)$ so that $\omega(\widehat{z}_i) = z_i$ when $\widehat{z}_i \notin \mathrm{Im}(\sigma)$. Combining with the definition of $\omega$ on $\mathrm{Im}(\sigma)$, we have successfully constructed a surjective map $\omega : [L] \to [K]$ satisfying $\omega(\widehat{z}_i) = z_i$ for all $i \in [n]$. The proof is complete. $\qquad\square$