# OpenReview forum: "Label consistency in overfitted generalized $k$-means"
_NeurIPS.cc/2021/Conference — NeurIPS 2021 Poster_

### Official Review · Reviewer_nGs4 · 2021-07-12

**Rating:** 6
**Confidence:** 4

**Summary:**

This paper considers the problem of label consistency in the generalized k-means problem in an overfit situation where the number of clusters $L$ used by a given clustering algorithm is more than the ground truth $K$.

The main contributions are:

1. Theoretical guarantees the above problem, under certain assumptions on the clustering algorithm, for both  exact and approximate label recovery. To the best of my knowledge, this is the first result in the context of the k-means problem.

2. The results are model-free, but strongly depend on the maximum or average distance of the points to true cluster center.



**Limitations And Societal Impact:**

Yes.

**Main Review:**

I found the paper decently written. The introduction is comprehensive, the setup and the learning problem are clear. I read all the proofs in the main body of the paper and the supplementary material, and everything looks correct and standard.

I have a few comments/questions (in a random order).

1.  Assumption 1:

a. In Assumption (1b) is it clear that both $K$ and $L$ can be associated with the same \kappa? Obviously $\kappa$ can be taken to be the maximum between those approximation constants.

b. You are claiming that some of your results hold with assumption (1b') which you claim ( in page 4) to be a relaxation. As far as I could follow, Theorems 1 and 2, in fact, are proved using Assumption (1a) and (1b'), and so in the statements of Theorem 1 and 2, I do not understand why you are referring to Assumption 1? Then, right before Subsection 2.2 you are mentioning that you can relax part (b) of Assumption 1. This is a bit confusing. More importantly, I do not understand in what sense (1b') is a relaxation of (1b). It seems as though that neither conditions follow from each other. For example, if we assume that (1b) holds, why this implies that (1b') is satisfied as well? Vice-versa, if (1b') holds, while it might seem that (1b) should hold, it does not becasue you won't be able to obtain the right hand side guarantee for the constant approximation for $L$. Am I missing something?

c. It would be much more clear if you provide a complete statement of Lemmas 1-3, mentioning which assumptions are needed? For example, in Lemma 1 you use Assumption (1b'), in Lemma 2 you use Assumptions (1a) and (1b'), etc.

2. Necessary conditions:  while the sufficient condition in Theorems 1 and 2 are nice, it is difficult to appreciate those results since it is not clear at this point whether whatever you obtained is anything close to what is necessary. What is the difficulty with deriving lower bounds (even simple/loose ones)?

3. Can you point out "typical" values of the parameters such as $\eta$, $\delta$, $epsilon$, etc.? If those are fixed, then fine, but what if those parameters depend on $K$ and $n$? For example, if $\delta$ is relatively small with n, then your bounds are sometimes not informative.

4. The sentence following equation 4 is vague. Can you add more details on how the Theorem shows whatever is written there?

5.  You are mentioning that condition (4) can be relaxed from "+4" to "+2". Unless I am missing something, this seems like a negligible improvement, and I do not see the point of mentioning this.

6. In Theorem 2, if $\epsilon$ is large enough then you can get to zero misclassification error using Theorem 1. Therefore, you can at least have a slightly better bound there.

7. The motivation, result, and the sentence following (7), surrounding Theorem 3 are not clear and vague. It would be great if you could clarify some of the details there.

8. You mention a few times in the paper that you proofs are non-trivial. I think that these statements should be removed.

9. In Theorem 2 you say "Then, for any c > 2". What is the benefit of have this $c$, and not just using $c=2$. This is the only condition that is needed for the proof, and this is the "best" value for $c$ both in terms of the misclassification error and the separation condition.

10. In some sense, the theorems statements seems incomplete in the sense that it is not clear what happens when the margin condition is reversed, namely, can you say something meaningful to what happens when $0<\delta<...$.

Conclusion:

While I do believe that the setup and the questions asked in this paper are interesting and important I find the theoretical results quite weak. Currently the paper presents some "upper bounds" which are difficult to appreciate due to luck of some kind of "lower bounds" to compare with. The dependency of Theorem 1 and 2 on the various parameters is quite strong and it is not clear at all whether this dependency is inherent. Furthermore, in terms of technical contribution, I found the proofs quite straightforward (certainly not non-trivial). Therefore, I my current recommendation is 5.

Typos:

8. In the proof of Theorem 1 it should be $+4\eta$ and not $+4\epsilon$.

9. When you bound the cardinality of $S_k$ in the proof of Theorem 2 it should be a strict inequality $<1$.

10. "We now extend Theorem 3" should be "We now extend Theorem 2".

11. "Have properties in Assumption 1"  should be "Have the properties in Assumption 1".

12. "value the cost function" should be "value of the cost function"..

**Time Spent Reviewing:**

48

---

> ### Author Response · Authors · 2021-08-10
> **Response to Reviewer nGs4**
>
> Thanks for the careful reading of our paper and for your detailed comments.
>
> 1. **Assumption 1**:
> **a.** $\kappa$ could depend on the number of clusters $K$ (or $L$) -- say in the case of k-means++ where $\kappa = O(\log K)$ -- but there are also constant approximation k-means algorithms where $\kappa = O(1)$ independent of $K$ (or $L$). As pointed out by Reviewer j782, ‘Clustering under approximation stability’ by Balcan, M. F., Blum, A., & Gupta, A. (JACM  2013) includes some examples.
>
> **b.** Theorems 1 and 2 hold under either (1b) or (1b’); however, we need (1b) for Theorem 3. There we cannot replace (1b) with (1b’). The reason why we decided to use Assumption 1 is to avoid less confusion by using a consistent set of assumptions for all the three results and commenting where things can be slightly relaxed. (Apparently, this has had the opposite effect!)
> Regarding the relaxation issue, you are correct that strictly speaking (1b') is not a relaxation of (1b).  Informally, though, we can think of (1b) in a sense as a relaxation of (1b’) since it only requires the $\kappa$-approximation to hold for the true number of clusters $K$ while (1b) requires it to hold for all  $L \ge K$. The added price we pay for this relaxation is adding montonicity to (1b’), which is a “negligible price’’; no reasonable algorithm would violate montonicity. We try to either remove the word “relaxation” in the revision or elaborate more.
>
> **c.** “... complete statement of Lemmas 1-3…” Good suggestion. We will implement this in the revision.
>
> 2. This was a great suggestion! We plan to add a result on it, in the revision. Please see our “General comments to reviewers” where we address the **necessary condition** (a.k.a. the **lower bound**) by providing an example leading to such a lower bound on the misclassification rate. There, we have given details of the example as well as a proof sketch.
> 3. $\eta, \delta$ and $\varepsilon$ refer to bounds on the key quantities controlling the hardness of the problem ($\eta$ and $\varepsilon$ are essentially controlling the same thing). Generally, there are no typical values for them and they depend on the problem. They are not assumed to be constants and they are allowed to vary. In some problems they might depend on $n$; that in itself does not render the results uninformative. What matters in the end is the ratio $\delta / \varepsilon$ not the absolute sizes of these parameters, and as we demonstrate with the **lower bound**, this ratio being large is fundamental to the recovery. That being said, we use the sub-Gaussian mixture model (Example 2 on P4) as an example on how $\delta$ and $\varepsilon$ scale in a typical stochastic model.
> 4. We will. When letting $L = K$, the surjective map $\omega$ in Definition 1 will be a bijection, that is, a permutation on $[K]$. Then, $\text{Miss} = 0$ means that there is a permutation of the estimated cluster labels that turn the estimated label vector $\widehat{z}$ equal to the true label vector $z$.
> 5. We agree. We will remove the comment.
> 6. The two theorems are based on bounding different notions of distance to the centers, i.e., maximum versus average distance. Ideally, in Theorem 2 we would keep $\varepsilon$ roughly equal to the left hand side, which is really like the variance of the data to the true centers. In Theorem 1, we have $\eta$, which ideally is (pushed to down to) the largest distance of the data points to the centers and it can be significantly larger than $\varepsilon$.
> 7. The motivation is twofold: First, the result gives a theoretical confirmation that as the number of clusters increases, the misclassification rate decreases.  As the theorem states, when $L$ increases, we can always construct “fake” centers to make $\varepsilon$ smaller, while $\delta$ stays the same as it is still the separation of true clusters.
> Second, it shows that a refinement of true labels can be achieved in the overfitting case when it is hard to recover true labels with the true number of clusters. Section 3 is devoted to fleshing out this idea by applying the result to various examples.
> 8. We will remove these.
> 9. In the proof, it is required to have $c$ strictly larger than 2. Also, having $c$ as a parameter relaxes the separation condition. If we state it with a fixed constant say $2.1$ and you can only verify the separation constraint with a slightly larger $c$, say $2.5$, then there is no result. Having it like what we have stated allows whatever constant you manage to get for the separation to carry over “softly” to the result, rather than putting a hard constraint on the separation condition.
> 10. This goes back to your point 2 on the **necessary condition** which we will provide in the revision as discussed earlier. See our “General comments to reviewers” for details.
>
> Thanks for pointing out the typos. We will take care of them in the revision.

---

> > ### Comment · Reviewer_nGs4 · 2021-08-25
> > **Lower bound**
> >
> > Thank you for the detailed feedback.
> >
> > Regarding the lower bound in your proof sketch (which is not simple to follow, but I do not have specific comments), can you summarize here your conclusion? Namely, write explicitly both your upper and lower bounds so that it will be easier to compare?

---

> > > ### Author Response · Authors · 2021-08-27
> > > **Summary of upper and lower bounds**
> > >
> > > > Regarding the lower bound in your proof sketch (which is not simple to follow, but I do not have specific comments),
> > >
> > > We have updated the figure as well as the proof sketch with more details which hopefully makes it easier to follow.  We also added the details of why applying the algorithms with $L=4$ clusters is guaranteed to recover a perfect refinement. Please see [the updated proof sketch](https://ibb.co/B6J5NFX).
> > >
> > > > ... can you summarize here your conclusion? Namely, write explicitly both your upper and lower bounds so that it will be easier to compare?
> > >
> > > Sure. Consider the case $K=2$ since our lower-bound example is for this case. (The upper bound already holds for general $K$ and the example can be generalized too.) Just for simplicity, we also consider 2-factor $k$-means algorithms.
> > >
> > > **The upper bound** (a corollary of Theorem 2): Consider a vector of true centers $\xi^* \in \mathcal X^2$ and target labels {$z_i$}. Let $\delta$ be the minimum separation between the centers in $\xi^*$ (or a lower bound on it) as in (3) in the paper. Assume that $\frac1n \sum_i \| x_i - \xi_{z_i}\|^2 \le \varepsilon^2$. Then, for every $\beta > 0$, there exists a constant $c_1(\beta, \pi_{\min})$ such that if $\delta/ \varepsilon \ge c_1(\beta,\pi_{\min})$, then, the missclassification rate of any 2-factor $k$-means algorithm to the target labels satisfies: Miss $\le \beta$.
> > >
> > > In the above, one can take $c_1(\beta,\pi_{\min}) = 6.3\sqrt{\max( \frac1{\pi_{\min}}, \frac{2}{\beta})}$ which is obtained by taking $\kappa = K = p = 2$ and $c=2.1$ in Theorem 2.
> > >
> > > **The lower bound:**
> > > There exists a family of 2-cluster datasets {$(x_i,z_i)$}, parametrized by true center separation $\delta$ and $\varepsilon = (\frac1n \sum_i \|x_i - \xi^*_{z_i}\|^2)^{1/2}$ with the following property:
> > > Given any constant $\beta \in (0,1/2)$, there exists a constant $c_2(\beta) > 0$, such that if $\delta / \varepsilon < c_2(\beta)$,
> > > then any 2-factor $k$-means approximation algorithm with $K=2$ clusters has misclassification rate  satisfying $1/2 - \beta \le \text{Miss} \le \frac12$. That is, the misclassification rate for $K=2$ can be made arbitrarily close to that of random guessing. Moreover, any 2-factor $k$-means approximation algorithm with $L=4$ clusters will recover a perfect refinement of the original clusters in the above setting.
> > >
> > > The true centers in the above statement are defined by {$(x_i,z_i)$} via $\xi^*_k = $ the mean of {$x_i: z_i = k$}.
> > >
> > > One can take $c_2(\beta) = \sin (\tan^{-1} (\sqrt{\beta/45}))$ in the statement of the lower bound. See the proof for details.

---

### Official Review · Reviewer_8yKF · 2021-07-15

**Rating:** 7
**Confidence:** 3

**Summary:**

The paper considers the problem of label recovery from data clustering. Generally speaking, these two tasks are unrelated: the label recovery is a supervised learning problem, where each data point has a unknonwn true label, and the goal is to infer the label for each data point; while the clustering problem is to group data together to minimize the generalized $k$-means cost (i.e., replace $\ell_2$-error with $\ell_p$-error in traditional $k$-means), which usually has nothing to do with labelings. Although the former task also induces a clustering (in which data points with the same label belongs to the same cluster), it can be a very bad solution for the later task, and vice versa. To distinguish, for a given data set, we say a clustering is a **true clustering** if it recovers the unknown labeling (up to permutations); while we say it is an **optimal $k$-means clustering** if it solves the second task, i.e., minimizes the generalized $k$-means cost function.

The paper shows that, when the *true clustering* satisfy certain regularity conditions, it can be recovered (up to "refinement") by (an $O(1)$ approximation to) the \emph{optimal $k$-means clustering}. The paper defines a **refinement** of labeling $z\in[K]^n$ to be another labeling $\tilde{z}\in[L]^n$ together with a mapping $\omega:[L]\mapsto[K]$, such that $\omega(\tilde{z}_i)=z_i$ for all $i\in[n]$. Intuitively one can think of this as breaking the *true clusters* further into multiple smaller clusters.

Per my understanding the main result can be summarized as follows. When the true clustering is itself a good clustering with respect to the generalized $K$-means cost (Theorem 1 and 2), or if the true clustering can be broken into $L>K$ smaller clusters which have small $L$-means cost (Theorem 3), then an $O(1)$-approximation for the optimal $L$-means clustering is also a good approximation for the true clustering, in the sense that it approximately induces a refinement of the true labeling.

**Main Review:**

**[Strength]**
The paper proposes an interesting metric of classification error in the overfitted case, which seems to be the first to do so (I'm not familiar with literatures on this particular problem, though). The paper also gave rigorous analysis on the conditions when a refinement can be recovered. I like the applications given in section 3: The paper shows that when the true clustering is induced by some smooth manifold, a not-to-large $L=O(\sqrt{n\log n})$ is sufficient to recover a refinement of the true labeling.

**[Weakness]**
I don't have time to check the proofs in detail, but the main result is intuitively unsurprising: the regularity condition is essentially assuming that the labeling induces a clustering that's good w.r.t. the $k$-means cost. Furthermore, the authors also assume the true clusters (or its refinement) are well-separated. With such assumptions it's really not surprising that a good $k$-means clustering also approximates the true clustering well. Furthermore, I'm wondering what's the dependence of $L$ in all the bounds obtained: when $L=n$ it trivially recover a refinement of the true labeling, even if the well-separation condition does not hold. It would be good to show that we don't need too large $L$ to achieve small labeling error. (I'm not sure if it's possible though, maybe it requires stronger assumptions). Besides, in practice the separation parameter $\delta$ is usually unknown and untrollable, while $L$ is something we can specify. So I believe it's better to at least have some experiment on realworld data sets showing the effect of $L$.

--------
# Acknowledgement of author response [Sep 2]
-------
I have read the author response and am satisfied with it. But I do think the authors should revise the paper in their post-rebuttal version: (1) Clarify the difference between the two clustering tasks involved: one is inferring the labels, and the other is minimizing certain cost function. (2) Discuss the relation between their assumption with the "distribution stability" mentioned by Reviewer wtkZ.

Overall, I have a positive feeling on their submission, and this has not changed after seeing their rebuttal, thus I'm increasing my score to 7. But I have to admit that I'm not familiar with the techniques or the literatures, so I don't have very high confidence in my score.



**Time Spent Reviewing:**

6

---

> ### Author Response · Authors · 2021-08-10
> **Response to Reviewer 8yKF**
>
> Thanks for your eloquent summary of our work and your comments. The first paragraph in your summary is in fact perfect! This is what we should have said in the introduction. With your permission, we will borrow from you and revise the introduction to reflect how you set up the problem.
>
> > … the main result is intuitively unsurprising: the regularity condition is essentially assuming that the labeling induces a clustering that's good w.r.t. the $k$-means cost. Furthermore, the authors also assume the true clusters (or its refinement) are well-separated. With such assumptions it's really not surprising that a good k-means clustering also approximates the true clustering well.
>
> Having the right intuition is different from having a proof. A rigorous result makes the intuition “precise”. Exactly how one should measure the separation between clusters? How big the separation needs to be? Relative to what? How should one measure the deviations to the centers? As you can see from Theorem 1 and 2, there are different ways with different consequences. Combined with the “Necessary Condition” result which we will add to the revision (as detailed in the general comments to the reviewers), our work shows that a single parameter like $\delta/\varepsilon$ is the fundamental quantity controlling the problem (nothing more and nothing less). This is not in itself that intuitive.
>
> The situation is even less intuitive for $L > K$. We provide the line-circle model and the circle-torus model as two examples where the clusters are “intuitively” well-separated but a k-means clustering (with $K$ clusters) would fail because cluster centers are overlapped (i.e., they are not separated from a k-means cost perspective with $K = 2$ as you suggest). Going to $L > 2$ in these cases, how exactly should we define the separation and between what centers, to get a useful result? We believe that Theorem 3 is not that intuitive. Note that the fake centers in that theorem need not all be separated and some pairs can have a zero distance.
>
> > Furthermore, I'm wondering what's the dependence of L in all the bounds obtained: when $L = n$ it trivially recover a refinement of the true labeling, even if the well-separation condition does not hold.
>
> The dependence is implicit. For example, in Theorem 3, as one increases $L$, satisfying separation (6) with carefully constructed centers becomes easier. We have a little discussion after Theorem 3 that generally, increasing $L$ allows you to make $\varepsilon$ smaller while maintaining $\delta$ roughly fixed, allowing the bound on the misclassification rate to go down. It is not clear whether it can be made more explicit than this, without assuming more about the data generation process.
>
> > It would be good to show that we don't need too large $L$ to achieve small labeling error.
>
> This is an interesting point and in general a hard problem. A lot depends on the model one assumes for the data and we are far from having a complete answer.  We have given examples in Section 3 bounding how large $L$ needs to be to achieve small labeling error with $L = o(n)$ which is already nontrivial. In the supplement we have provided empirical evidence that suggests in some cases one can expect $L = O(1)$ whereas in others $L = O(\sqrt{n \log n})$ might be unavoidable.
>
> > So I believe it's better to at least have some experiment on realworld data sets showing the effect of $L$.
>
> This is an interesting suggestion. We will consider adding some real world examples in the revision.

---

> ### Comment · Reviewer_8yKF · 2021-08-26
> **Acknowledgement of author response**
>
> I have read the author response and am satisfied with it. I'm increasing my score to 7.

---

### Official Review · Reviewer_wtkZ · 2021-07-15

**Rating:** 7
**Confidence:** 4

**Summary:**

This paper considers the problem of recovering (sub)clusters with respect to the k-means objective, assuming that the centers satisfy certain separations. Specifically, the authors show that when running an algorithm with more than k centers, the subclusters are consistent with the target k-clustering.

**Ethical Concerns:**

Not necessary for this submission.

**Limitations And Societal Impact:**

Not necessary for this submission.

**Main Review:**

I think that the authors raise an interesting question. While recovering a target clustering $C_k$ may be hard for a given value of k, recovering a clustering $C_L$ with $L>k$ centers such that the target clustering $C_k$ could be recovered from $C_L$ is an interesting way of modeling data compression in an (exact) recovery setting in a way that, for example, coresets would not be able to.

I initially misunderstood what the author's assumptions were. Now it is clearer that their separation results are related to (but slightly stronger than) distribution stability introduced by Awasthi Blum and Sheffet in FOCS 2010. Showing certain recovery notions under these assumptions has been studied in Cohen-Addad Schwiegelshohn (FOCS 2017) and for the related notion of spectral separability introduced by Kumar Kannan (FOCS 2010) and Awasthi Sheffet (APPROX 2012), but I believe that the results here are not obviously weaker than what is achieved in those reference and the idea of recovering a solution with more centers has certainly not been studied in the aforementioned resutls.

Now that I also understood the lower bound (and the assumptions), I also think that the example given by the authors where their algorithm provides a separation is a compelling one. Overall, after the rebuttal my impression is a lot more positive and I believe the paper could be accepted.

**Time Spent Reviewing:**

4

---

> ### Author Response · Authors · 2021-08-10
> **Response to Reviewer wtkZ**
>
> Thanks for your comments. Please see our “General comments to reviewers” for clarifications on some points regarding our setting and motivation which perhaps were not clear based on our original presentation. We believe that hopefully this will clear up some points of departure between your view on the problem and ours. We believe both viewpoints are valid and interesting and both would contribute to a better understanding of clustering problems.
>
> > I am not sure if the authors are aware of an increasing literature in "beyond worst case analysis" for k-clustering objectives including k-means. I would very much recommend that the authors relate their results to that line of research.
>
> Thanks for pointing this out. We are aware of the trend which is along the line of the “smoothed analysis” for analyzing computational complexity. We will add comments in the revision discussing the relation. Based on our understanding, the area studies algorithm performance in cases where meaningful structures exist in data rather than considering all the cases, hence being restricted by the worst case. Often, a stochastic model for the data is assumed and high probability bounds (or bounds in expectation) are obtained. In other words, one analyzes the average or typical case, rather than the worst case.
>
> Our work is perfectly aligned with this line of research. Our recovery result is based on the separation of true cluster centers, indicating that we assume “meaningful clusters in the data” to be discovered. We are not concerned with approximating the optimal k-means objective value well for all inputs. It is enough to have a good  approximation only on the instances of interest as pointed out in “General comments to reviewers”.
>
> Our work follows another trend in analyzing statistical algorithms of separating the deterministic part of analysis from the stochastic part. Theorem 2, for example, posits deterministic conditions sufficient for recovery and Example 1 verifies that they hold with high probability in a common stochastic model. As such we very much care about the average/typical case when a stochastic model exists.
>
> > … specialized algorithms have proven to recover approximately or even optimal clusterings, under separation assumptions significantly weaker than presented here.
>
> It would have helped if you provided specific examples of these types of results. Without knowing the specifics, it is hard to accurately comment here. But we would like to counter with a few points:
> - We will provide a reverse-direction result in the revision showing that our separation condition cannot be improved in general beyond constants. (Please see “General comments to reviewers”). The example we give is also not super exotic and variations on it could easily occur in practice. If there are results with better guarantees, they must put a lot more assumptions on the data-generating process and most likely assume a restricted class of true cluster labelings.
> - Even if ‘specialized algorithms’ under carefully constructed circumstances have better guarantees, these guarantees often fall apart with slight deviations from the assumed models for the data. There is value in providing results under minimal assumptions, both on the data generating process and the specificity of the algorithm which is what we aimed to do here. One cannot anticipate what interesting cases future researchers will encounter and we hope that they can still apply our results to get some meaningful bounds for their problems. In fact, our work was motivated by colleagues who wanted to use k-means clustering to cluster the states of a Markov process. They could not find any existing results that can be applied in this case.
> - Even though specialized algorithms with, say, near-optimal theoretical guarantees (under specific setups) are nice to have, they are not the ones that are most often used in practice. What gets used is the likes of kmeans++ and it is good to have guarantees for these algorithms.
>
> In short, our results are complementary to the types of results you suggest and we believe that there is value in both lines of work.
>
> > These algorithms have bad worst-case guarantee but work extremely well in many recovery regimes, which illustrates why a separation based on the worst case approximation ratio of the algorithm is not particularly meaningful.
>
> We did not fully understand this comment. Our separation condition is not based on considering the worst-case instance. (Please see point 3 of the “General comments to reviewers”). But we need the algorithm to  “sort of” minimize the k-means objective function, because we want to understand what can be achieved by k-means type clustering (not k-clustering in general). If an algorithm has a bad worst-case approximation ratio, but can approximate the k-means objective function well over a subclass of instances and you are willing to assume that your data is within that subclass, you can apply all our results in that case too. In other words, the algorithm should provide a \kappa-approximation to the k-means objective only “on the dataset that you are considering.” We will add this comment to the revision.
>
> > Unfortunately, the authors only prove the part concerning itself with overfitting, without showing that recovery for k-clustering is hard (the authors claim that this is the case, but do not show it).
>
> There are such examples already included in Section 3 where k-means would fail to recover labels but overfitting can produce refinement of the true clusters. See for example the line-circle model where the k-means objective function will be the same (i.e., constant) for all partitions of the data into $K=2$ clusters. This *proves* that 2-clustering with the k-means objective function fails in this case. We have pointed this out in the paper, but will make it more clear in the revision. Another example is that of concentric spheres which we will add.
>
> > Even Local Search, which is among recovery algorithms the one with the best worst-case guarantees, performs a lot better in recovery regimes than the bounds here indicate.
>
> Please see our general comment about “worst-case analysis” and the “necessity of our separation condition” in the general comments to all reviewers. Regarding “performs a lot better in recovery regimes … “, we are not aware of any results in the overfitted regimes we consider here.
>
> > I was also interested in seeing how the authors were going to prove that overfitting helps recover (parts of the) correct k-clustering in the case where recovery for k-clustering is hard.
>
> Please see our Theorem 3 and its applications in Section 3. Specifically, the use of the “fake centers”, which is only possible under overfitting, is what allows us to prove recovery bounds not otherwise possible.

---

> > ### Comment · Reviewer_wtkZ · 2021-08-10
> > **Worst Case Approximation Ratio**
> >
> > I am sorry for misunderstanding your assumptions then. But this was not clear and could have been written better. I will re-read in light of what $\kappa$ is supposed to mean. It would have been better if you had made this comment a bit earlier as it would have given us more time to discuss.
> >
> > So if I understand correctly, you are saying that any $\kappa$ approximation on that input would be enough to recover the optimum as well? In this case your work is closely aligned to that of Blum, Balcan and Gupta, "Approximate Clustering without the Approximation". I am checking if I now understand the definition at least. I also believe that your condition is related to what Awasthi, Blum, and Sheffet in FOCS 2010 coined distribution stability (the two stability notions are connected).
> >
> > Whether or not you wish to cite these papers and follow up results is up to you. But consider that relating yours to these notions would rule out misunderstandings which I had. Given that these were the papers I had in mind, which do define the separation in terms of a worst-case approximation ratio, I think that comparing your work to previous stability notions would have made it a lot easier on the reader.
> >
> > As for my comment regarding the lower bound on an instance that is only separable when adding more centers? Did you actually evaluate all $2^n/2$ clusterings for Figure 1? How did you obtain this lower bound on the cost of any clustering? Generally, I do not like proofs by plot and if it is indeed that simple, then a few comments would have helped a great deal. Maybe, now that I understand what your Theorem 1 is actually proving, I no longer consider this part the most important contribution of your paper. But I did so initially, and even now it is not clear to me how a proof for the separation would work.

---

> > > ### Author Response · Authors · 2021-08-10
> > > **RE: Worst Case Approximation Ratio**
> > >
> > > > I am sorry for misunderstanding your assumptions then. But this was not clear and could have been written better ...
> > >
> > > Thanks for being open and for bringing up the issue in the first place. We agree. This  was not clear in our writing and we plan to rectify it in the revision.
> > >
> > > > So if I understand correctly, you are saying that any  approximation on that input would be enough to recover the optimum as well? ...
> > >
> > > Yes, that is correct. See, for example, L286 for how $\kappa$ comes up in the proof. It is only needed to deal with the present input {$x_i$}.
> > >
> > > > Whether or not you wish to cite these papers and follow up results is up to you ...
> > >
> > > Thanks for pointing out the connections to those papers. We will relate our work to those in the revision.
> > >
> > > > As for my comment regarding the lower bound on an instance that is only separable when adding more centers? Did you actually evaluate all $2^n/2$ clusterings for Figure 1? ...
> > >
> > > We have added a link to [a proof sketch](https://ibb.co/SPLrn7z) at the end of the general comments. There is no need to evaluate all the clusterings directly. The idea is to use Theorem 2 to show that the Missclassification rate will be small relative to a  clustering different from the intended target, hence the Miss. rate to the target will be large. Theorem 2 is flexible enough with the choice of the centers to allow for such an argument.

---

> > > > ### Author Response · Authors · 2021-09-01
> > > > **Distribution Stability**
> > > >
> > > > Thanks again for pointing to the work of Awasthi, Blum and Sheffet in FOCS 2010. Here is an update: Our separation condition is indeed closely related to their distribution stability. Roughly speaking "distribution stability" plus the following property implies our separation:
> > > >
> > > > - (D1) For every pair of distinct clusters $C_k$ and $C_\ell$ with centers $\xi^*_k$ and $\xi^*_\ell$, there is a point $x \in C_\ell$ such that  $\|| x - \xi^*_k\|| \le \||\xi^*_\ell - \xi^*_k\||$.
> > > >
> > > > That is, every cluster $C_\ell$ has points which are closer than $\xi_\ell$ to the centers of other clusters. This property is quite mild and one expects it to hold with high probability if the distribution of the points have positive density w.r.t. to the (full-dimensional) Lebesgue measure in a ball around the center.
> > > >
> > > > The above seems to suggest that distribution stability is slightly weaker than our separation. However, in the presence of (D1) we can also significantly relax "distribution stability" to arrive at our separation. See [this link for details](https://ibb.co/mtvPhhk). In this sense, these two notions are closely related but not directly comparable (neither is weaker than the other in general).

---

### Official Review · Reviewer_j782 · 2021-07-16

**Rating:** 7
**Confidence:** 4

**Summary:**

The paper proves exact and approximate recovery guarantees for the generalized $k$-means problem under some (natural) assumptions on the data points. These guarantees hold for any constant factor approximation algorithms for the problem and their results are valid not only for the case when $k$ is correctly specified, but also when $k$ is over estimated, which make their results more relevant for practical applications.

**Limitations And Societal Impact:**

As mentioned by the authors, the work is mainly theoretical in nature and does not have any potential negative societal impact.

In page 2, just below part (b) of the assumption, it is stated that reference [2], which is $k$-means++ achieves constant approximation. But $k$-means++ is an $O(\log k)$ approximation algorithm. Please correct this. There are (of course) several other constant factor approximation algorithms which you can choose to cite. I would also suggest looking at the paper "Clustering under Approximation Stability" by Balcan, Blum, and Gupta (JACM 2013). They also study the recovery error for k-means and k-medians problems and I think it is quite related to your work. There has also been some prior work on overfitted clustering, which you study in the paper, for instance: Wei (NIPS 2016) and Makarychev, Reddy, Shan (NeurIPS 2020).

Some minor typos: In the line just above definition 1, misclassification is misspelled as "misclassiciation". Towards the end of the second paragraph of section 3.3, "in the in the Supplementary Material".

**Main Review:**

The paper is well written, clear, and interesting to read. The "refinement" idea they introduce for studying label consistency in the overfitted case is very natural and quite interesting. I am surprised it hasn't been studied in any prior work. The generalized $k$-means problem studied by the paper encompasses the classic $k$-means and $k$-median problems, which are obviously of interest to members of the NeurIPS community.  Most prior work has focused on the value achieved by the optimization function but in practice, what is more relevant is the recovery error i.e. how close are the labels assigned to the true labels, which is studied in this paper.

**Time Spent Reviewing:**

5-6 hours

---

> ### Author Response · Authors · 2021-08-10
> **Response to Reviewer j782**
>
> Thanks for your positive feedback and pointing out the references.
>
> Regarding the approximation factor for k-means++, thanks for catching the error. We will correct it in the revision and add your suggested references for other approximation algorithms.
>
> Also thanks for pointing out the connection to Wei (NIPS 2016) and Makarychev, Reddy, Shan (NeurIPS 2020). We will discuss them in the revision. The main difference between our work and theirs is that they consider the approximation of the objective function relative to its optimal value in the overfitting case, while we do not care about the objective function value. As pointed out by Reviewer 8yKF in their first paragraph, we focus on recovering the unknown true labels which can be unrelated to the labels obtained by optimizing the objective function.

---

> ### Comment · Reviewer_j782 · 2021-08-25
> **Acknowledgment of author response**
>
> Hi, I just wanted to acknowledge that I have read the author responses and the reviews by the other reviewers. My positive impression of the paper hasn't changed by much after doing so. Thank you.

---

### Author Response · Authors · 2021-08-10
**General Comments to Reviewers**

We thank all the reviewers for their time and effort. Besides responses to individual comments, we would like to make some general comments that address some of the negative points raised by the reviewers. In the revision, we will try to rectify misunderstandings in the current version:

1. We are not considering $k$-clustering in general. We are concerned with what can be achieved by $k$-means clustering, i.e., by (approximately) optimizing a $k$-means type objective function.

2. For us, the true cluster labels are coming from elsewhere using external considerations and need not have anything to do with the k-means objective function. Hence for us there is no “optimal clustering”. In other words, the optimal clustering is whatever the labels {$z_i$} suggest, which may be in conflict with what k-means objective thinks should be the optimal clustering based on the structure of the data. We believe that this is the most reasonable assumption in practice when dealing with label consistency and this is also in agreement with **Reviewer 8yKF**’s view on the subject as pointed out so eloquently in their first paragraph of the summary. Therefore, our results are provided in the context where the true labels {$z_i$} are allowed to be arbitrary. Perhaps the early parts of Section 1 (the introduction) could have been interpreted differently and we plan to revise it to make it more consistent with our setting.

3. We are not concerned with the worst-case approximation properties of the algorithms. The $\kappa$ in our results only needs to be the approximation factor of the algorithm on the “specific dataset” that you care about. If an algorithm has an $O(n)$ approximation factor on all the data except a subclass (or even an instance) for which it is $O(1)$ and you only care about that subclass (or instance), then our results apply to your problem. We will make this point more clear in the revision.

4. Since our goal is to study $k$-means clustering, we need to maintain some form of connection to the k-means objective function, hence the idea of looking at constant-factor approximation algorithms.

Furthermore, we would like to address the **necessary condition** issue raised by **Reviewer nGs4**. First, thank you for bringing up this interesting point. We will add a result to the revision addressing this exact question. In short, the result shows that, in general, our separation condition cannot be improved beyond constants.

To achieve the above, we demonstrate a dataset with 2 true clusters, true center separation $\delta$ and $(\frac1n \sum_i ||x_i - \xi^*_{z_i}||^2)^{1/2} = \varepsilon$ for true cluster labels {$z_i$}, with the following property: Given any constant $\beta \in (0,1/2)$, there exists a constant $c(\beta) > 0$, such that if $\delta / \varepsilon < c(\beta)$, then any 2-factor $k$-means approximation algorithm with $K=2$ clusters has misclassification rate  bounded as $|\text{Miss} - 1/2| \le \beta$, that is, the misclassification rate can be made arbitrarily close to that of random guessing.

This dataset also has the additional property that an $L=4$ clustering will recover a perfect refinement of the original clusters by any 2-factor $k$-means algorithm.

---
A picture of the setup can be found [at this link](https://ibb.co/tCcJmCk) with details and proof sketch [available here](https://ibb.co/SPLrn7z).

---

> ### Author Response · Authors · 2021-08-27
> **An updated statement of the lower bound**
>
> In response to Reviewer nGs4, below is an updated statement of the lower bound (i.e., the necessity of the condition) with a [more detailed proof sketch with improved constants](https://ibb.co/B6J5NFX):
>
> **The lower bound:**
> There exists a family of 2-cluster datasets {$(x_i,z_i)$}, parametrized by true center separation $\delta$ and $\varepsilon = (\frac1n \sum_i \|x_i - \xi^*_{z_i}\|^2)^{1/2}$ with the following property:
> Given any constant $\beta \in (0,1/2)$, there exists a constant $c_2(\beta) > 0$, such that if $\delta / \varepsilon < c_2(\beta)$,
> then any 2-factor $k$-means approximation algorithm with $K=2$ clusters has misclassification rate  satisfying $\frac12- \beta \le \text{Miss} \le \frac12$. That is, the misclassification rate for $K=2$ can be made arbitrarily close to that of random guessing. Moreover, any 2-factor $k$-means approximation algorithm with $L=4$ clusters will recover a perfect refinement of the original clusters in the above setting.
>
> The true centers in the above statement are defined by {$(x_i,z_i)$} via $\xi^*_k = $ the mean of {$x_i: z_i = k$}.
>
> One can take $c_2(\beta) = \sin (\tan^{-1} (\sqrt{\beta/45}))$ in the statement of the lower bound. See the proof for details.

---

### Decision · Program_Chairs · 2021-09-27

**Decision:**

Accept (Poster)

**Comment:**

This paper shows that in certain settings, overparameterized k-means can recovery a refinement of the true clusters even when recovering the true clusters themselves is impossible. The reviewers agree that this is a novel observation that broadens the applicability of k-means and our understanding of when performance guarantees can be obtained. The paper is also timely and adds to the recent literature on overspecified models.

The reviewers also point out several weaknesses of this work:
- The assumptions, in particular the separation condition, appear strong. While the authors promise to add a lower bound in their response, the lower bound does not reflect dependence on the dimension.
- The conclusion seems unsurprising under their assumptions, and the analysis involves standard techniques.

The paper can be made stronger by incorporating several suggestions from the reviewers, including
- Adding a lower bound.
- Clarifying the difference between the two clustering tasks involved.
- Discussion of the relation between their assumption with "distribution stability" (mentioned by Reviewer wtkZ), as well as with the assumptions made in prior work on Gaussian mixture models.